# Multi-Sensor Fusion for Activity Recognition—A Survey

**DOI:** 10.3390/s19173808

**Published:** 2019-09-03

**Authors:** Antonio A. Aguileta, Ramon F. Brena, Oscar Mayora, Erik Molino-Minero-Re, Luis A. Trejo

**Affiliations:** 1Tecnologico de Monterrey, Av. Eugenio Garza Sada 2501 Sur, Monterrey, NL 64849, Mexico; 2Facultad de Matemáticas, Universidad Autónoma de Yucatán, Anillo Periférico Norte, Tablaje Cat. 13615, Colonia Chuburná Hidalgo Inn, Mérida, Yucatan 97110, Mexico; 3Fandazione Bruno Kessler Foundation, 38123 Trento, Italy; 4Instituto de Investigaciones en Matemáticas Aplicadas y en Sistemas—Sede Mérida, Unidad Académica de Ciencias y Tecnología de la UNAM en Yucatán, Universidad Nacional Autónoma de México, Sierra Papacal, Yucatan 97302, Mexico; 5Tecnologico de Monterrey, School of Engineering and Sciences, Carretera al Lago de Guadalupe Km. 3.5, Atizapán de Zaragoza 52926, Mexico

**Keywords:** multi-sensor fusion, activity recognition, survey

## Abstract

In Ambient Intelligence (AmI), the activity a user is engaged in is an essential part of the context, so its recognition is of paramount importance for applications in areas like sports, medicine, personal safety, and so forth. The concurrent use of multiple sensors for recognition of human activities in AmI is a good practice because the information missed by one sensor can sometimes be provided by the others and many works have shown an accuracy improvement compared to single sensors. However, there are many different ways of integrating the information of each sensor and almost every author reporting sensor fusion for activity recognition uses a different variant or combination of fusion methods, so the need for clear guidelines and generalizations in sensor data integration seems evident. In this survey we review, following a classification, the many fusion methods for information acquired from sensors that have been proposed in the literature for activity recognition; we examine their relative merits, either as they are reported and sometimes even replicated and a comparison of these methods is made, as well as an assessment of the trends in the area.

## 1. Introduction

The use of context in modern computer applications is what differentiates them from older ones because the context (the place, time, situation, etc.) makes it possible to give more flexibility so that the application adapts to the changing needs of users [1].

One of the most critical aspects of context is the identification of the activity the user is engaged in; for instance, the needs of a user when she is sleeping are completely different from the ones of the same subject when is commuting. This explains why the automated recognition of users’ activity has been an important research area in recent years [2]. Recognition of these activities can help deliver proactive and personalized services in different applications [3].

Human Activity Recognition (HAR) based on sensors has received much attention in recent years due to the availability of advanced technologies (such as IoT) and its important role in several applications (such as health, fitness monitoring, personal biometric signature, urban computing, assistive technology, elder-care, indoor localization and navigation) [4]. The recognized activities could be “simple” activities defining the physical state of a user, for instance, walking, biking, sitting, running, climbing stairs, or more “complex” ones defining a higher level intention of the user, such as shopping, attending a meeting, having lunch or commuting [5].

The HAR researchers have made significant progress in recent years through the use of machine learning techniques [6,7], sometimes using single-sensor data [8,9,10]. Some commonly used learning techniques are the logistic regression (LR) [11], decision tree (CART) [12], the Random Forest Classifier (RFC) [13], Naive Bayes (NB) [14], Support Vector Machine (SVM) [15], K-nearest neighbors (KNN) [16] and the Artificial Neural Networks (ANN) [17]. Among the sensors commonly used there are accelerometers, gyroscopes and magnetometers [18,19]. However, the use of a single sensor in the HAR task has been unreliable because most sensors have limited information due to sensor deprivation, limited spatial coverage, occlusion, imprecision and uncertainty [20].

In order to address the issues of using one sensor and improve the performance (measured mainly by accuracy, recall, sensitivity and specificity, see Section 3.6) of the recognition, researchers have proposed methods of multi-sensor data fusion. Using multiple sensors for recognizing human activities makes sense because the information missed by one sensor can sometimes be provided by the others and also the imprecision of a single sensor can often be compensated by other similar ones. Many works have shown an accuracy improvement compared to single sensor [21]. Now, there is a wide variety of methods for combining the information acquired from several similar or different sensors and there are active research areas called “Sensor Fusion”, “Information Fusion” and similar ones, which have dedicated journals, conferences, and so forth. Even when restricting our attention to multi-sensor fusion in the context of HAR, there are hundreds of specific works having used many variants of combinations of fusion methods and each author claims to have achieved better results than others, so the need for putting some order in this area seems evident.

For instance, we have found methods that fuse the features extracted from the sensor data, such as Aggregation [22,23]. Methods that fuse the decisions of the classifiers associated with each sensor, such as Bagging [24], Voting [25], Adaboost [26], Multi-view stacking [27], a system of hierarchical classifiers proposed by He et al. [28], to mention some. “Mixed methods” fuse the characteristics of sensor data and the decisions of the classifiers associated with the sensors, such as a method based on a sensor selection algorithm and a hierarchical classifier (MBSSAHC) [29]. So, although these fusion methods have in different ways improved the performance of activity recognition and have overcome to a certain degree the problems of using a single sensor, they have been piled chaotically one on top of the previous ones, so that there is no order, which is a hurdle both for studying the area, as well as for identifying the specific situations where some methods could be better fitted than others.

In the view of these considerations, in this work we have made a significant effort in identifying the main families of fusion methods for HAR and we have developed a systematic comparison of them, which is the main substance of this survey.

Later in this paper, we will present these techniques in a structured way, providing an ontological classification to guide the relative placement of one method with respect to the others.

In short, this paper presents a survey about multi-sensor fusion methods in the context of HAR, with the aim of identifying areas of research and open research gaps. This survey focuses on trends on this topic, paying particular attention to the relationship among the characteristics of the sensors being combined, the fusion methods, the classification methods, the metrics for assessing performance and last but not least, the datasets, which we suspect should have some traits making some fusion methods more suitable than others.

The rest of this paper is organized as follows. In Section 2, we compare this survey with other surveys. In Section 3, we present the background, which shows the main concepts and methods of this area. In Section 4, we present the methodology. In Section 5, we show the fusion methods. In Section 6, we discuss the findings and limitations of the study. Finally, we present the conclusions in Section 7.

## 2. Incremental Contribution Respect to Previous Surveys

In Table 1, we compare other HAR surveys that have been published (such as the Gravina et al. review [20], Chen et al. survey [30] and Shivappa et al. survey [31]) with this survey. In this table, we note that one of the main differences between these surveys and ours is the type of sensors of interest. We do not limit ourselves to a specific type of sensor; we are interested in all kinds of sensors (external and wearable [32]) because we want a broad perspective of fusion methods. Another difference is the way of explaining the fusion methods. Whereas these surveys explain fusion methods in a general way, we are more interested in explaining these methods in detail. a detailed explanation of these methods can provide insight into the main ideas behind them and, with this vision, researchers can better understand and better choose some of them.

Also, unlike the other surveys, we are interested in contrasting the performance reached by the fusion of heterogeneous sensors (sensors of different classes, such as accelerometers and gyroscopes) and the fusion of homogeneous sensors (sensors of the same class). We are also interested in contrasting the fusion methods that manually extract the features (for instance, extracting statistical features by hand—mean, standard deviation, to mention a few) and fusion methods that automatically obtain the features (for example, using Convolutional neuronal networks [33]). Likewise, we focus on comparing the performance achieved by the methods that mix at least two fusion methods (“Mixed fusion”) and the methods that use a fusion method (“Unmixed fusion”). We consider these three crucial comparisons due to the types of sensors to be fused, the way of extracting the features during fusion and the number of fusion methods that can be mixed can affect the performance of the recognition of the activity, as we will see in Section 6.

In addition to these differences, other specific differences are presented per survey. The survey by Gravina et al. [20] focuses on data fusion in HAR domains, emotion recognition and general health. In particular, they categorize the literature in the fusion at the data level, at the feature level and at the decision level (typical categorization) [34]. Also, they compare the literature, identifying the design parameters (such as window size, fusion selection method, to mention some) and the fusion characteristics (such as communication load, processing complexity, to mention a few), at these fusion levels. Our survey differs mainly from Gravina in that we add the classification “Mixed fusion” and we do not focus our attention on the fusion parameters, nor on the characteristics of the fusion.

Chen et al. [30] present the state of the art of the techniques that combine human activity data that come from two types of sensors—vision (depth cameras) sensors and inertial sensors. They classify these techniques in the fusion at the level of data, at the level of features and at the decision level. Also, they discuss the fusion parameters. Our survey differs mainly from Chen in that we add the “Mixed fusion” classification and do not focus our attention on the fusion parameters.

Finally, the main differences between Shivappa et al. [31] and our survey are the classification scheme and the concepts to be recognized. Concerning the classification scheme, Shivappa uses seven categories to classify the fusion methods, whereas we use four categories. However, we agree on three categories with Shivappa. In the concepts to be recognized, Shivappa focuses on speech recognition, tracking, identification of people (biometrics), recognition of emotions and analysis of the scene of the meeting (analysis of the human activity in rooms meeting) and we focus mainly on the recognition of physical activity. Another difference with Shivappa is that we do not focus our attention on fusion parameters.

## 3. Background

In this section, we present the central concepts used by our work, to unify the terminology. Also, these notions are explained in sufficient detail so that a non-expert can get a quick understanding of the topic.

### 3.1. Human Activity Recognition

In recent years there has been increased attention to research on HAR [35,36,37,38] for several reasons. One of them is that it can lead to costs savings in the management of common diseases, such as diabetes, heart and lung diseases, as well as mental illness, which will cost $47 trillion each year from 2030 [39]. As physical activity is key in the prevention and/or treatment of these illnesses [40,41,42,43,44], research on HAR is an enabler to better performance monitoring and diagnosing [45,46,47]. For example, Bernal et al. [48] propose a framework for monitoring and assisting a user to perform a multi-step rehabilitation procedure. Kerr et al. [49] present an approach to recognizing sedentary behavior. Rad et al. [50] put forth a framework to the automatic Stereotypical Motor Movements detection.

Another use case for HAR is to identify falls of elder people [51]. The number of old people is increasing in the world [52] Indeed, in 25% of the cases, the population of adults over 65 will suffer at least one fall, every year, with consequences ranging from bone rupture to death [53,54]. As examples of studies on the care of the elderly, we list Alam’s research [55] that proposes a framework for quantifying the functional, behavioral and cognitive health of the elderly. Also, the detection of falls of elderly people has been studied by Gjoreski et al. [56], Li et al. [57] and Cheng et al. [58].

We find another motivation for research on HAR in sports. For example, Wei et al. [59] propose a scheme for sports motion evaluation. Ahmadi et al. [60] present a method to assess all of an athlete’s activities in an outdoor training environment. Also, Ghasemzadeh et al. [61] come up with a golf swing training system that provides feedback on the quality of movements and Ghasemzadeh et al. [62] system evaluates and gives feedback for the swing of baseball players. Other applications of HAR are in Ambient-Assisted Living [63], marketing [64], surveillance [65] and more.

Finally, recognizing human activities is of great interest because they can give relevant information about the situation and context of a given environment to these applications [2].

#### Definition of Human Activity Recognition

According to Lara et al. [32], the Human Activity Recognition (HAR) problem can be defined as—Given a set $S=\{S_{0},..,S_{k-1}\}$ of *k* time series, each one from a definite measured characteristic and all defined within time lapse $I=[t_{\alpha ,},t_{\beta }]$, the objective is to look a temporal partition $\langle I_{0},\ldots ,I_{r-1}\rangle$ of *I*, according to the data in *S* and a set of labels which represent the activity performed during each interval $I_{j}$ (e.g., sitting, walking, etc.). This definition implies that time lapses $I_{j}$ are consecutive, they are not empty, they do not overlap and in such a way that ${⋃}_{j=0}^{r-1}I_{j}=I$. In this definition, the activities are not concurrent. The next definition supports this concurrency with some insignificant errors [32].

Definition 2 (Relaxed HAR problem)—Given (1) a set $W=\{W_{0},..,W_{m-1}\}$ of *m* windows of the same size, totally or partially tagged and such that each $W_{i}$ includes a set of time series $S_{i}=\left\{S_{i,0},..,S_{k-1}\right\}$ from each of the *k* measured characteristics and (2) a set $A=\{a_{0},..,a_{n-1}\}$ of activity labels, the aim is to meet a mapping function $f:S_{i}\to A$ that can be assessed for all possible values of Si, such that $f(S_{i})$ is as similar as possible to the actual activity carried out during $W_{i}$ [32].

### 3.2. Sensors in Human Activity Recognition

In the area of recognition of human activity, both external sensors and portable sensors have been widely used [10]. External sensors are installed near the subject to be studied and portable sensors are transported by the user [32]. Video cameras, microphones, motion sensors, depth cameras, RFID tags and switches are example of external sensors. Accelerometers, gyroscopes, magnetometers are instances of wearable sensors [10].

About external sensors, they could be expensive due to the number of sensors that must be purchased to increase system coverage [32]. For example, cameras and proximity sensors have limited coverage according to their specifications [66].

Concerning such portable sensors, they consume little energy, provide large amounts of data in open environments and can be purchased at low cost [67]. Also, these sensors are less likely to generate privacy problems compared to external sensors, such as video cameras or microphones [10]. The central problem with wearable sensors is their elevated level of intrusion [32].

### 3.3. Machine Learning Techniques Used in HAR

In this section, we present the techniques of automatic learning classification commonly used by researchers in the HAR field. This is not an extensive review of the ML field; the reader can consult a general ML book as the ones from Bishop [6] and from Mitchell [68].

Logistic Regression (LR)[11] is a statistical technique for finding the relationship between a dependent variable and one or two independent variables, in order to predict the occurrence of an event by modeling the influence of the variables related to that event or to estimate the value of the dependent one.

Decision Tree (DT) is a classification technique where a decision is taken by following the test nodes of the tree from the root to a leaf, which defines the class label for a given sample [68]. To achieve a good generalization and depending on the values of the available features, this tree tries to deduce a division of training data. This division of the tree nodes is formed according to the maximum information increase and the leaf nodes are associated with the class labels. Each of these nodes on the route tests some characteristics of this sample. The classification and regression trees [69] and ID3/C4.5 [70] are examples of algorithms for setting up decision trees.

Random Forest (RF) [13] method constructs a set of random variations of classification trees, based on a feature vector. Each of these trees generates a decision that this method uses to produce the final decision [71]. Researchers use this method extensively because of its simplicity [72].

k-Nearest Neighbors (KNN) [16] method is based on the idea that examples with similar characteristics maintain the proximity between them. Due to this proximity, it is possible to classify an unknown instance by observing the class of the nearest instances. Then, KNN determines the class of some example by identifying the most frequent class tag of the nearest k examples. The value of K is generally defined by a validation or cross-validation set.

Naïve Bayes (NB) [14] is a method based on the combined probability of the features (*x* vector) given a truth label (*y*). This method is defined as $p\left(x_{1},\ldots ,x_{n}|y\right)=\prod_{j=1}^{n}p(x_{i}|y)$, where $x=(x_{i},\ldots ,x_{n})$ and $x_{i}$ is conditionally independent given *y*. So, the class label *c* of one unknown example is assigned according to the class with the highest probability given the observed data ($c=argmax_{c}P\left(C=c|x_{1},\ldots ,x_{n}\right)$). Given a problem with *K* classes $C_{1},\ldots ,C_{K}$ with probabilities $P\left(C_{1}\right),\ldots ,P\left(C_{K}\right)$, $P\left(C=c\right)|x_{1},\ldots ,x_{n})=P\left(C=c\right)P\left(x_{1}|C=c\right)\ldots P\left(x_{n}|C=c\right)$.

Support Vector Machine (SVM) [15] uses the idea of maximizing the margins of a hyperplane (optimal hyperplane) that divides two types of data. This hyperplane and its ρ separation margin can be formulated as $w^{T}x+b=0$ [73] and as $\frac{2}{∥w∥}$, respectively. Also, this hyperplane can be optimized by minimizing ρ with respect to *x* and *b*, $min\frac{1}{2}{∥w∥}^{2}$, $s.t.y_{i}\left(w^{T}x_{i}+b\right)\geq 1,i=1,\ldots ,n$. With the kernel function technique, which is generally used to separate data that are not linearly separable [74], the optimal classifier is defined as $f\left(x\right)=\sum_{i=1}^{n}\alpha_{i}y_{i}K\left(x_{i},x\right)+b$, where α is the optimal Lagrange multiplier and $K(x_{i},x)$ is the kernel function. Among the kernel functions that stand out are the radial base ($exp(-y∥x_{i}-x{∥}^{2}),y>0$) and the sigmoidal ($tanh(x_{1}^{T}x+c)$).

Artificial Neural Network (ANN) [17] consists of a group of connected “neurons” with weights, inspired in the structure of a brain. a detailed description of ANN is beyond the scope of this paper; please see a detailed description in a ML book like the Mitchell one [68].

Convolutional Neural Networks (CNNs) [33] are a type of ANN. These CNN perform convolutional operations (matrix multiplication) on filters (matrix) to extract characteristics on a given dataset. These features are more straightforward in shallow layers and more complex in deep layers.

Recurrent Neural Networks (RNNs) are also a type of ANN. These RNNs are comprised of units that use the pre-activation information to produce the next information activation and the corresponding predictions. The objective of these RNNs is using the previous information to achieve predictions about data representing sequences over time.

Long-short-term Memory Networks (LSTMs) are a type of RNN. These LSTMs aim to remember the dependencies between the data representing time sequences. These dependencies could be remembered for a prolonged period. To remember this dependence over a long period, these LSTMs use gates to update the status of the units—writing (entry door), reading (exit door) or reset (forgetting door).

Multilayer Perceptron Neural Network (MLP) is a neural network in which many input nodes are joined with weights associated with various output nodes. The output of the network can be estimated from the addition function $o_{i}=\phi \left(\sum_{i}W_{i}x_{i}\right)$, where $W_{i}$ is the weight used to fit the input $x_{i}$ and ϕ is the activation function [75]. MLP infers the classification error through a backward propagation algorithm and attempts to find the weights to minimize that error.

Radial Base Function Neural Network (RBF) [75] works with the RBF as an activation function. For *N* hidden neurons, the activation function is $f\left(x\right)=\sum_{i=1}^{N}W_{i}\varphi (\|x-c\|)$, where *c* is the central vector for the neuron *i* and φ is a function of the nucleus.

### 3.4. Activity Recognition Workflow

The typical HAR workflow is a sequence of steps illustrated in Figure 1.

In the first step, the raw data is obtained from whichever sensors are used, such as accelerometers, gyroscopes, pressure sensors (for body movements and applied forces), skin/chest electrodes (for electrocardiogram, ECG), electromyogram (EMG), galvanic skin response (GSR) and electrical impedance plethysmography (EIP)), microphones (for voice, ambient and heart sounds), scalp-placed electrodes for electroencephalogram (EEG). Raw data are sampled, generating a multivariable time series. Notice that each sensor could have a different sampling rate, as well as varied limitations of the power supply, space restriction, and so forth. Thus, achieving synchronization between multimode sensor data presents technical difficulties, such as the time difference between the sensors, the corruption of unprocessed sensor data caused by physical activity, sensor malfunction or electromagnetic interference [76]. Some techniques to sample the raw data are fixed rate, variable rate, adaptive sampling, compressed sensing and sensor bit-resolution tuning [77,78].

In the processing step, different algorithms are applied to the raw data coming from sensors to address the aforementioned problems and leave the data ready for the extraction of features. For example, acceleration and gyroscope signal filtering usually include calibration, unit conversion, normalization, resampling, synchronization or signal level fusion [79]. Physiological signals, such as electrooculography (EOG), generally require preprocessing algorithms to eliminate noise or eliminate baseline drift [80]. The challenge for these algorithms is that they must retain the raw data properties that are important in the discrimination of human activities [76].

In the segmentation step, the processed data obtained from the previous step is split into segments of adequate length. This segmentation is not an easy task because the human fluently performs actions and there is no clear delimitation between activities [76]. However, there are several methods to overcome to some extent this difficulty, such as the sliding window [81], energy-based segmentation [82], rest-position segmentation [83], the use of one sensor modality to segment data of a sensor of another modality [84] and the use of external context sources [76].

The fourth step extracts the characteristics of the segmented data from the previous level and organizes them into vectors that together form the space of the characteristics. Examples of these characteristics are the mean, variance or kurtosis (statistical features). The mel-frequency cepstral coefficients or energy in specific frequency bands (Frequency-domain features) [85]. Features extracted from a 3D skeleton generated by body sensors (Body model features) [86]. Encoded duration, frequency and co-occurrences of data (expressive feature) [5,87].

Also, in this step, the task of selecting features is performed because the reduction of them is essential to reduce computational needs. Because the task of manually choosing such features is complicated, several techniques have been developed to automate this selection [76], so that these can be categorized into wrapper [88], filter [89] or hybrid [90] methods. Convolutional Neural Networks (CNNs) have also been used for feature selection [91].

In the training step, the inference algorithms are trained with the features extracted in the fourth step and the actual labels (“ground truth”). During training, the parameters of these algorithms are learned by reducing the classification error [76]. Among the methods of inference that are usually used are the k-NN (k-Nearest-Neighbor) [16], Support Vector Machines (SVM) [15], Hidden Markov Models (HMM) [92], Artificial Neural Networks (ANN) [17], Decision Tree Classifiers [12], Logistic Regression [11], Random Forest Classifier (RFC) [13] and the Naive Bayesian approach [14].

In the *classification* step, the model trained in the previous step is used to predict activities (mapping feature vectors with class labels) with a given score. The final classification can be done in many ways, such as choosing the highest score and letting the application choose how to use the scores [76].

### 3.5. Multi-Sensor Data Fusion on HAR

Multisensor fusion had its origins in the 1970s in the United States Navy as a technique to improve the accuracy of motion detection of Soviet ships [93]. Nowadays, various applications use this idea for applications such as the supervision of complex machinery, medical diagnostics, robotics, video and image processing and intelligent buildings [94].

Multisensor fusion techniques refers to the combination of the features extracted from data of different modalities or the decisions generated from these characteristics by classification algorithms [95]. The objective of sensor fusion is to achieve better accuracy and better inferences than a single sensor [21]. So, sensor fusion has the following advantages compared to the use of a single sensor [96]:Enhanced signal to noise ratio—the merging of various streams of sensor data decreases the influences of noise.Diminished ambiguity and uncertainty—the use of data from different sources reduces the ambiguity of output.Improved confidence—the data generated by a single sensor are generally unreliable.Increased robustness and reliability, as the use of several similar sensors provides redundancy, which raises the fault tolerance of the system in the case of sensor failure.Robustness against interference—raising the dimensionality of the measuring space (for example, measuring the heart frequency using an electrocardiogram (ECG) and photoplethysmogram (PPG) sensors) notably improves robustness against environmental interference.Enhanced resolution, precision and discrimination—when numerous independent measures of the same attribute are merged, the granularity of the resulting value is finer than in the case of a single sensor.Independent features can be combined with prior knowledge of the target application domain in order to increase the robustness against the interference of data sources.

Regarding the level of abstraction of data processing, Multi-sensor fusion can be divided in three main categories—data-level fusion, feature-level fusion and decision-level fusion [34]. These categories are defined as:

Data-level fusion: It is generally assumed that the combination of multiple homogeneous sources of raw data will help to achieve more precise, informative and synthetic fused data than the separate sources [97]. Studies on data-level fusion are mainly concerned with the design and implementation of noise elimination, feature extraction, data classification and data compression [98].

Feature-level fusion: Feature sets extracted from multiple data sources (generated from different sensor nodes or by a single node with multiple physical sensors) can be fused to create a new high-dimensional feature vector [30]. Also, at this level of fusion, machine learning and pattern recognition, depending on the type of application, will be applied to vectors with multidimensional characteristics that can then be combined to form vectors of joint characteristics from which the classification is carried out [99].

Decision-level fusion: The decision-level fusion is the process of selecting (or generating) a class hypothesis or decision from the set of local hypotheses generated by individual sensors [100].

These levels of fusion takes its place in the activity recognition fusion and, in doing so, they configure an extended version of it (see Figure 2). In Figure 2, the merging at the data level occupies the second position because the raw data of several sensors feed this level. The fusion at the feature level is located between the step of extraction and selection of the characteristics and the training step since this training requires the features extracted from the sensors. The decision-level merger occurs both in the training stage and in the classification stage because the decisions of some classifiers are combined to make a final decision.

### 3.6. Performance Metrics

The performance for a particular method can be organized in a confusion matrix [76]. The rows of a confusion matrix show the number of instances in each actual activity class, whereas the columns show the number of instances for each predicted activity class. The following values can be obtained from the confusion matrix in a binary classification problem:

True Positives (TP): The number of positive instances that were classified as positive.

True Negatives (TN): The number of negative instances that were classified as negative.

False Positives (FP): The number of negative instances that were classified as positive.

False Negatives (FN): The number of positive instances that were classified as negative.

Accuracy, precision, recall and F-measure (also F1-score or F-score) are the metrics most commonly used in HAR [101]; as well as, the specificity. Below, we present these metrics:

Accuracy is the most standard measure to summarize the general classification performance for all classes and it is defined as follows:(1)$$\mathrm{Accuracy}=\frac{TP+TN}{TP+TN+FP+FN}$$

Precision, often referred to as positive predictive value, is the ratio of correctly classified positive instances to the total number of instances classified as positive:(2)$$\mathrm{Precision}=\frac{TP}{TP+FP}$$

Recall, also called sensitivity or true positive rate, is the ratio of correctly classified positive instances to the total number of positive instances:(3)$$\mathrm{Recall}=\frac{TP}{TP+FN}$$

F-measure combines precision and recall in a single value:(4)$$\text{F-measure}=2\cdot \frac{Precision\cdot Recall}{Precision+Recall}$$

Specificity measures the proportion of negatives that are genuinely negative:(5)$$\mathrm{Specificity}=\frac{TN}{TN+FP}$$

Although defined for binary classification, these metrics can be generalized for a problem with *n* classes.

## 4. Methodology

To find relevant studies on the subject of this work in a structured and replicable way, in this survey, we rely on a systematic mapping study [102]. This method provides an extensive overview of a field of research and a structure for a given research topic [103]. Also, this method offers a guide for an exhaustive analysis of the primary studies of a particular theme and to identify and classify the findings of this review [104]. The process to carry out a mapping study is described in the following subsections.

### Identification and Selection of Sources

For this survey, we picked the Scopus database. This database contains a broad range of scientific writing and provides a reliable and friendly search engine and a diversity of tools for exporting results [105]. Figure 3 presents the overall workflow of the search procedure we followed in this mapping study.

Next, we defined the search string by combining the logical operators “AND” and “OR” with the terms obtained from the research question. Table 2 presents the resulting search string. This string shows that most of these terms are related to the recognition of human activities through the use of multiple sensors.

Once the search source was picked out and the search string was established, we reduced the selection of primary studies by applying the inclusion criteria (CI) and the exclusion criteria (CE). The inclusion and exclusion criteria are presented in Table 3.

After determining the inclusion and exclusion criteria, we executed the search string in the Scopus database. We analyzed the search results with respect to the recognition of human activities. Regarding the selection procedure, we admitted the studies according to the inclusion and exclusion criteria. We examined the titles, abstracts and keywords of all article search results. Also, we reviewed the entire document. Table 4 shows the number of resulting documents and the number of relevant documents selected. These results include papers published up to 2018.

## 5. Fusion Methods

In this section, we classify the different fusion methods found in the literature. This classification is guided by the merging categories presented in Section 3.5—data-level fusion, feature-level fusion and decision-level fusion.

### 5.1. Methods Used to Fuse Data at the Data Level

In this section, we consider the methods that fit the merge category at the data level (see Section 3.5). Therefore, the methods classified here share the characteristic that their final predictions are made by trained classifiers with the combined raw data of the sensors.

#### 5.1.1. Raw Data Aggregation

In the category of merging at the data level, we show the raw data aggregation (RDA) method that consists of concatenating the raw data of all the sensors, extracting the features of them and training a classification model with these features. The procedure followed for this concatenation is to first segment the raw data of each sensor according to its sampling frequency and then concatenate these segments taking into account the time [50].

#### 5.1.2. Time-lagged Similarity Features

Time-lagged Similarity Features (TLSF) [106] is a method that fuse at the level of data. In this method, the signal strength measurements are processed for pairs of devices to calculate time-lagged similarity features, based on raw signal strength measurements or derived location measurements using location fingerprinting [107]. Formally, these time-lagged features are computed for a pair of devices *a*, *b* as a vector $v_{a,b}$, where each entry is associated with a certain time lag i∈[-z,…,z] in seconds, where *z* defines the range of time lags. Hence, the length of $v_{a,b}$ is 2z+1 and for each time lag *i*, $v_{a,b}$ then holds a feature value, which indicates the similarity of the measurements from *a* and *b*, when shifting those of *b* by a time lag *i*. Each feature value is computed over a time window of size *w* over $m_{a}$ and $m_{b}$. Here, $m_{a_{t}}$ is the measurements for time step *t* for a device *a*.

### 5.2. Methods Used to Fuse Data at the Feature Level

In this section, we consider the methods that fit the merge category at the characteristic level (see Section 3.5). Therefore, the methods classified here share the characteristic that their final predictions are made by trained classifiers with the combined features of the sensor data.

#### 5.2.1. Feature Aggregation

In the category of fusion at the level of features, we present the method of Feature Aggregation (FA), which consists of concatenating the characteristics extracted from all the sensors (vector of features) and training a unique classification model with this vector [22,23,108,109,110,111,112,113,114,115]. Sometimes, FA is complemented by Principal Component Analysis (PCA) [116] to reduce the dimension of the feature vector.

#### 5.2.2. Temporal Fusion

Temporal Fusion (TF) [48,50] is another method used to fuse features. This method consists of automatically extracting features from raw data of the sensors using a CNN and fusing these features using a LSTM.

#### 5.2.3. Feature Combination Technique

Feature Combination (FC) [117] method selects a group of characteristics that, together, achieve the best overall performance of a neural network by measuring the impact of fixed features in this network. FC, in addition to a neural network, uses the clamping technique [118]. The following steps describe this method:

Given S={}, $F=\{f_{1},f_{2},...,f_{N}\}$ and g(), a set of selected features, a set of N features and the generalized performance of the network, respectively.
Calculate the feature importance for all features $f_{i}$ in *F* using $Im\left(f_{i}\right)=1-\frac{g(F|f_{i}={\bar{f}}_{i})}{g(F)}$.Select the feature fs in the feature space *F* which has the maximum impact $f_{s}=max_{f_{i}\in F}Im\left(fi\right)$.If and only if $g\left(S\cup f_{s}\right)\geq g\left(S\right)$, then update *S* and *F* using $S=S\cup \{F_{s}\}$ and $F=F\backslash \{F_{s}\}$Repeat steps 2 and 3 for N-1 times.

#### 5.2.4. Distributed Random Projection and Joint Sparse Representation Approach

In Distributed Random Projection (DRP) [119], the random projections (RP), an almost optimal measurement scheme, of the signal measurements are made in each source node, only considering the temporal correlation of the sensor readings.

Let $a^{j}\left(t\right)=\left(x^{j}\left(t\right),y^{j}\left(t\right),z^{j}\left(t\right),\theta^{j}\left(t\right),\varphi^{j}\left(t\right)\right)\in R^{5}$ indicates the five measurements afforded by sensor *j* at time *t*, where j=1,…,J and each *j* wearable sensor has a 3-axis accelerometer (x,y,z) and a 2-axis gyroscope (θ, φ). Also, $v^{j}={\left[a^{j}\left(1\right),a^{j}\left(2\right),\ldots ,a^{j}\left(h\right)\right]}^{T}\in R^{5h}$ represents an action segment of length *h* by node *j*. In addition, let $\Phi_{j}$ be the random projection matrix (*M* x *N*) for each sensor *j* and ${\tilde{v}}^{j}=\Phi_{j}v^{j}$ a vector after RP.

Each sensor *j* sends this vector ${\tilde{v}}^{j}$ to the base station (sink). This base station gather random projection vectors of *J* sensors and aggregates them as $\tilde{v}={\left[{\tilde{v}}^{1},\ldots ,{\tilde{v}}^{J}\right]}^{T}=\Phi v$, where $\Phi \in R^{M_{J}xN_{j}}$ is a matrix of diagonal blocks that was created by matrices of random projection of *J* sensors. In addition, the dictionary $V={\left[V^{1^{T}},\ldots ,V^{J^{T}}\right]}^{T}$ is built, where each $V^{j}$ is constructed with the training samples of the corresponding jth sensor. Considering that all the sparse representation vectors are the same, $\tilde{v}=\Phi V\beta +\epsilon$. The Joint Sparse Representation (JSR) [119] can be depicted by ${\tilde{v}}_{test}^{1}=\Phi_{1}V^{1}\beta^{1}+\epsilon_{1},\ldots ,{\tilde{v}}_{test}^{J}=\Phi_{J}V^{J}\beta^{J}+\epsilon_{J}$.

#### 5.2.5. SVM-Based Multisensor Fusion Algorithm

The SVM-based multisensor fusion (SVMBMF) algorithm [120] combines the data at the feature level. This algorithm consists of first extracting the time and frequency domain features of the sensors. Within the characteristics of the time domain are the mean value, the standard deviation, 10th, 25th, 50th, 75th, 90th percentiles and the correlation between the vector magnitudes. The frequency-domain features are the spectral analysis, including the frequency, spectral energy [121] and entropy [122]. Second, this algorithm performs a two-step feature selection process. The first step consists of a statistical analysis of the distribution of these features in order to have features whose distributions have the least overlap. The second step seeks to eliminate redundant features using the minimal-redundancy- maximal-relevance (mRMR) heuristic [89]. The mRMR approach measures the relevance and redundancy of the aspirant characteristics with the target class based on mutual information and choose a “promising” subset of features that have the greatest relevance and smallest redundancy. Finally, this algorithm combines the resulting characteristics that are introduced in the SVM classifier.

### 5.3. Methods Used to Fuse Data at the Decision Level

In this section, we consider the methods that fit the merger category at the decision level [100] (see Section 3.5). Therefore, the methods classified here share the property that their final decision is constructed with the outputs of several classifiers.

#### 5.3.1. Bagging

A method used to fuse the decisions of the base classifiers is Bagging [24]. This method uses the same learning technique with different subsets that were extracted from a given dataset. This extraction occurs with a replacement of the samples of this dataset. Each of these subsets was introduced in an instance of the learning technique. The prediction of each of these instances provides a vote for the final classification.

#### 5.3.2. Lightweight Bagging Ensemble Learning

Lightweight Bagging Ensemble Learning (LBEL) [123] is an expansion of the bagging algorithm. This extension consists of adding an inference engine based on the expression tree to help guide the process of recognition of activities in real time. LBEL bases in the BESTree selection algorithm [124] and Decision Tree (DT) as a base classifier. Formally, let *x* be a sample and $m_{i}$ (i=1…k) be a set of basic classification algorithms linked with the probability distributions $m_{i}\left(x,c_{j}\right)$ for each class label $c_{j},j=1..n$. The output of the final classifier set y(x) for instance *x* can be expressed as:$$y\left(x\right)=\underset{}{argmax}\sum_{i=1}^{k}w_{i}m_{i}\left(x,c_{j}\right)$$
where $w_{i}$ is the weight of base classifier $m_{i}$. LBEL uses ensemble learning approaches as subjacent methodologies for determining ideal weights for each base classification algorithm, given the hierarchical method that consists in the recognition of micro activity recognition and combining this with a semantic knowledge base (SKB) and location context for higher-level activity recognition.

#### 5.3.3. Soft Margin Multiple Kernel Learning

Since the definition of multiple kernel learning (MKL) is a sum of the weighted kernels, each trained in each sensor, this merging is done at the decision level. This MKL refers to a linear combination of some base kernels, such as RBF or linear cores [125]. A variant of this method is the soft margin MKL (SMMKL) [126] that makes SVM robust by entering the slack variables. This variant uses all the possible information and avoids remaining only with one of the sensors. Also, this variant is the counterpart of $L_{1}$MKL [127] that can be seen as a hard margin MKL, which selects the combination of a subset of base kernels that minimize the objective function and discard any other information (sensor).

#### 5.3.4. A-stack

A-stack [128] is a method that combines decisions of multiple classifiers. This method contain one base learner for each sensor and a meta learner. Each base learner is trained with the information of some sensor. The prediction scores of each base learner is combined in a vector. Finally, the meta learner is trained with this scores vector for the final predictions.

#### 5.3.5. Voting

Voting (Vot) is a method used to merge decisions of different classifiers [25]. In this method, the classifiers make their predictions that turn into votes. Based on these votes, the final prediction is made following a majority vote policy. Moreover, there is a variant of the way of voting, which is named weighted voting. In this type of voting, the classifiers are penalized according to its performance (accuracy or some other metric) by assigning to them a weight. In this way, the final classification is done by adding the weighted votes of the classifiers and choosing the class that reached the highest score.

#### 5.3.6. Adaboost

AdaBoost [26] is also a method that combines the decisions of the classifiers. Like Bagging technique, AdaBoost uses the same classifier with different subsets of a given dataset. However, this method focuses on the interactive training of weak instances of this classifier (poor precision). After the first training of some instance of these classifiers, where the examples of some subset of the dataset were assigned with the same weight, the weight of the examples that were not learned with precision increases. The idea behind the increase is that the next instance of the classifier pays more attention to these examples. This increase in weights produces a new subset of the dataset from which a new instance of the classifier is trained, and so on. In the end, the predictions of each instance of the classifier are taken into consideration in a weighted vote for the final classification. These weights are proportional to the accuracies achieved by each instance of the classifier.

#### 5.3.7. Multi-view Staking

Multi-view Staking (MulVS) [27] is based on multi-view learning [129,130] and stacked generalization [131]. This method consists of training one first-level learner for each view (sensors) and combining their outputs (class label, class predictions) using stacked generalization [131]. The final decision is performed by training of a meta-level learner with these combined outputs.

#### 5.3.8. Hierarchical Method

The hierarchical method combines the decision of different classifiers organized into several levels. Because this method uses the results of different classifiers in the task of classification, it is considered within the fusion at the decision level.

The Method Based on a Sensor Selection Algorithm and a Hierarchical Classifier (MBSSAHC) [29,132] follows a hierarchical structure of two levels. In the first level, this method trains the first classifier with the features extracted from the accelerometer data of a master node. Based on the results of this first classifier (class distribution) and expert knowledge (the distinctive capacity of a subset of sensors to distinguish distinct activities), this approach chooses a subset *S* of *K* sensors (nodes) different from the master node (in this case K=4 accelerometers ). Each node of the *S* subset sends its features to a fusion module. This fusion module constructs a vector *V* combining the features of the selected sensors and the information produced by the first classifier (the distribution of classes). In the second level, the final classification is produced using the second classifier that receives the vector *V*.

ubiMonitor—Intelligent Fusion (ubiMonitorIF) [133] is a hierarchical approach that uses three accelerometers and consists mainly of a Stationary detector, a Posture detector and a Kinematics detector. The stationary detector, the first level, uses a classifier (such as CART) to detect if a user is stationary or not. This classifier is fed with the harmonic mean and variance of the gravity acceleration from the three accelerometers. In the second level, for stationary users, the posture is detected using CART as the classifier and the gravity accelerations from the accelerometers as the features. In the third level, for non-stationary users, the user’s movement type is inferred using a binary decision tree classification algorithm based on SVM [133] with a set of descriptive features for the recognition of kinematic activities. This algorithm distinguishes several activities at different levels of abstraction. Sort the activities as a binary decision tree. Each node in the tree is a binary classifier (such as SVM). The upper node represents the highest level of abstraction of the activities. The nodes below represent the lowest levels of abstraction.

Hierarchical Weighted Classifier (HWC) [134] is a model that combines the decisions at activity level and sensor level. This model consists of three levels of decision making. The first level, called level of activity or class, is responsible for the discrimination of activities or classes. To achieve this discrimination, this level uses *M* by *N* base classifiers ($C_{mn},\forall m=1,\ldots ,M,n=1,\ldots ,N$), where *M* is the number of sensors and *N* is the number of classes (activities). These classifiers apply a binary classification strategy of one against the rest. The second level of classification, called sensor level, is configured with M sensor classifiers ($S_{m},\forall m=1,\ldots ,M$). This sensor classifiers are not machine learning algorithms but decision-making frameworks. Each sensor classifier have *N* basic classifiers (one per class), whose decisions are fused through an activity-dependent weighting design. The last layer, the network level, is responsible for the weighting and aggregation of the decisions provided by each sensor classifier, finally providing the identified activity or class. The weights used at the network level depend on the classification capabilities of each individual sensor classifier.

#### 5.3.9. Product Method

Product (Prod) method combines the probabilities of the classes predicted by the classifiers. Therefore, this method conforms to the fusion category at the decision level. The following formula defines this method:$$prediction_{i}^{j}=max_{K}\left\{\frac{1}{p{\left(C_{k}\right)}^{J-1}}\prod_{j=1}^{J}{\left({\hat{P}}_{ik}^{\left(j\right)}\right)}^{wj}\right\}$$
where $prediction_{i}^{j}$ is the prediction of the classifier j trained with the input $x_{i}$, ${\hat{P}}_{ik}^{\left(j\right)}$ is the posterior probability that $x_{i}$ belongs to class *k* and $w_{j}$ is the weight for classifier *j*.

#### 5.3.10. Sum Technique

The sum technique combines the probabilities of the classes predicted by the classifiers using the sum operation. This technique selects the class with the highest average probability. Therefore, this method conforms to the fusion category at the decision level. The following formula defines this method:$$prediction_{i}^{j}=max_{K}\left\{\frac{1}{J}\sum_{j=1}^{J}{\left({\hat{P}}_{ik}^{\left(j\right)}\right)}^{wj}\right\}$$
where $prediction_{i}^{j}$ is the prediction of the classifier j trained with the input $x_{i}$, ${\hat{P}}_{ik}^{\left(j\right)}$ is the posterior probability that $x_{i}$ belongs to class *k* and $w_{j}$ is the weight for classifier *j*.

#### 5.3.11. Maximum Method

The Maximum (Max) method is a decision-level merger strategy that decides the results according to the most reliable classifier. This technique selects the class with the highest probability from all classifiers. The following equation defines this method:$$prediction_{i}^{j}=max_{K}\left\{\frac{max_{J}{\left({\hat{P}}_{ik}^{\left(j\right)}\right)}^{wj}}{\sum_{k=1}^{K}max_{J}{\left({\hat{P}}_{ik}^{\left(j\right)}\right)}^{wj}}\right\}$$
where $prediction_{i}^{j}$ is the prediction of the classifier j trained with the input $x_{i}$, ${\hat{P}}_{ik}^{\left(j\right)}$ is the posterior probability that $x_{i}$ belongs to class *k* and $w_{j}$ is the weight for classifier *j*.

#### 5.3.12. Minimum Method

The Minimum (Min) method is a decision-level merger strategy. This technique selects the class that achieves the least objection for all classifiers. The following equation defines this method:$$prediction_{i}^{j}=max_{K}\left\{\frac{min_{J}{\left({\hat{P}}_{ik}^{\left(j\right)}\right)}^{wj}}{\sum_{k=1}^{K}min_{J}{\left({\hat{P}}_{ik}^{\left(j\right)}\right)}^{wj}}\right\}$$
where $prediction_{i}^{j}$ is the prediction of the classifier j trained with the input $x_{i}$, ${\hat{P}}_{ik}^{\left(j\right)}$ is the posterior probability that $x_{i}$ belongs to class *k* and $w_{j}$ is the weight for classifier *j*.

#### 5.3.13. Ranking Method

The Ranking (Ran) method selects the class with the highest rank. This rank is obtained by converting the probability ${\hat{P}}_{ik}^{\left(j\right)}$ into a rank. The values of these ranges fluctuate between 1 and *K*. Therefore, this method conforms to the fusion category at the decision level. The following formula defines this method:$$prediction_{i}^{j}=max_{K}\sum_{j=1}^{J}w_{j}rank_{ik}^{\left(j\right)}$$
where $prediction_{i}^{j}$ is the prediction of the classifier j trained with the input $x_{i}$, ${\hat{P}}_{ik}^{\left(j\right)}$ is the posterior probability that $x_{i}$ belongs to class *k* and $w_{j}$ is the weight for classifier *j*.

#### 5.3.14. Weighted Average

The weighted average (WA) method is a decision-level fusion strategy that bases its final decision on the sum of the weighted probabilities according to the following equation:$$prediction_{i}^{j}=max_{K}\sum_{j=1}^{J}w_{j}{\hat{P}}_{ik}^{\left(j\right)}$$
where $prediction_{i}^{j}$ is the prediction of the classifier j trained with the input $x_{i}$, ${\hat{P}}_{ik}^{\left(j\right)}$ is the posterior probability that $x_{i}$ belongs to class *k* and $w_{j}$ is the weight for classifier *j*.

#### 5.3.15. Classification Model for Multi-Sensor Data Fusion

Classification model for Multi-Sensor Data Fusion (CMMSDF) [135] is considered into the decision-level fusion because combine the decision of the symbolic information extracted form sensors. In this model, the data of each sensor is processed to get the essential information; for example, “ml01” for the level of movement. This basic information is fed into the system, which issues the answers. In the issuance of responses, only some of the processes are used to obtain the basic information. These processes are ordered, selected and used by the proposed model.

The proposed model is divided into three steps—In step 1 the selection of features is made, ordering the basic symbols that are compared in each type of activity. Step 2 compares the basic symbol obtained at a given moment with the database, which registers the previous symbols and serves in the training processes. Then, a table of the results of the comparison is created. Step 3 performs the analysis of the type of activity, which indicates whether each type of activity should discover more basic symbols or can guarantee its result without finding other basic symbols.

#### 5.3.16. Markov Fusion Networks

Markov Fusion Networks (MFN) [136] is a method used to combine decisions of various classifiers. The method combines temporal series of probability distributions of the classifiers. The combination is achieved by the following equations:

Here, *I* is the number of classes, *M* is the number of classifiers, *T* is the number of steps, $x_{m,t}\in {\left[0,1\right]}^{I}$ is the probability distribution of classifier *m* at time step $t\in \zeta_{m}$ where m∈1,...,M, $\sum_{i=1}^{I}x_{m,i,t}=1$ and $\zeta_{m}$ is the set of available probability distributions.

$p\left(Y,X_{1},\ldots ,X_{m}\right)=\frac{1}{Z}exp\left(-\frac{1}{2}\left(\Psi +\Phi +\Xi \right)\right)$ is the probability density function of the final estimation $Y\in {\left[0,1\right]}^{IxT}$ and the predictions of the classifier $X_{m}\in {\left[0,1\right]}^{IxT}$, where Z normalizes the probability to one.

$\Psi =\sum_{m=1}^{M}\Psi_{m}=\sum_{m=1}^{M}\sum_{i=1}^{I}\sum_{t\in \zeta }k_{m,t}{\left(x_{m,i,t}-y_{i,t}\right)}^{2}$ is the data potential, where $K\in R_{+}^{M,T}$ qualifies the reliability of the classifier *m* in the time step *t*.

$\Phi =\sum_{t=1}^{T}\sum_{i=l}^{I}\sum_{\hat{t}\in N\left(t\right)}w_{min(t,\hat{t})}{\left(y_{i,t}-y_{i,\hat{t}}\right)}^{2}$ is the smoothness potential, which models the Markov chain, where $w\in R_{+}^{T-1}$ ponders the difference between two neighboring nodes and N(t) returns the set of contiguous nodes.

$\Xi =u\cdot \sum_{t=l}^{T}\left({\left(1-\sum_{i=1}^{I}y_{i,t}\right)}^{2}+\sum_{i=1}^{I}1_{\left[0>y_{i,t}\right]}\cdot y_{it}^{2}\right)$ is the distribution potential, which ensures that the resulting estimate fits the laws of probability theory, where the parameter *u* ponders the pertinence of the potential and $1_{\left[0>y_{i,t}\right]}$ takes the value one in case *y* is negative.

#### 5.3.17. Genetic Algorithm-Based Classifier Ensemble Optimization Method

The method of classifier optimization based on genetic algorithms (GABCEO) [137] combines the result of the measurement level of different classifiers for each activity class to form the assembly of these classifiers. Because it combines these outputs, this method belongs to the decision level fusion category.

This method uses a Genetic Algorithm (GA) to optimize the measurement level output in terms of weighted feature vectors of classifiers. These weighted characteristics vectors of the classifiers are defined from their training performance for each class, which point out the chance that the values of the input sensor belong to the class. Also, these weighted feature vectors of all the learning algorithms are group into GA to infer the activity rules optimized for the final verdict on the activity class tag.

The architecture of this method consists of four main elements—(1) Data preprocessing, to draw the sensory data as an observation vector for the input of the classifier, (2) base classifier for the Activity Recognition (AR), to give details concerning the classifiers applied with the chosen configuration of parameters, (3) an apprentice of ensemble of the classifier based on GA, to optimize the vectors of weighted characteristics of multiple classifiers and (4) phase of recognition, to infer the activities carried out.

#### 5.3.18. Genetic Algorithm-Based Classifiers Fusion

Genetic Algorithm-Based Classifiers Fusion (GABCF) [138] approach consists of the following steps. First, the method receives raw data from various sensors. For *n* sensors, the raw input of the approach is defined as $x_{i},y_{i}$, where $x=S_{1},S_{2},...S_{n}$ y *y* is the result of *K* potential activities. The raw entries are preprocessed then using the weighted moving average (WMA). WMA is a strategy employed to soften the signal using $At=w_{1}A_{t}+w_{2}A_{t-1}$, where *A* is the signal at time *t*. Next, the data is scaled to the range [0 1]. Feature set *F* is extracted—mean, standard deviation (STD), maximum, minimum, median, mode, kurtosis, skewness, intensity, difference, root-mean-square (RMS), energy, entropy and key coefficient. *F* is entered into the feature selection process using the feature combination (FC) technique resulting in a feature set *S*. *S* is used in a classifier and passed through a multiple-classification block that yields class posterior probability P(j). Finally, the classifiers are merged and the fusion weights are determined using a genetic algorithm to produce the final prediction.

#### 5.3.19. Adaptive Weighted Logarithmic Opinion Pools

In Adaptive Weighted Logarithmic Opinion Pools (WLOGP) [139], the individual later likelihood $p_{j}\left(w_{i}|x\right),\left(j=1,2,\ldots ,n\right)$ are used to qualify the membership of the combined class as follows: $w=argmax\sum_{j=1}^{n}a_{j}p_{j}\left(w_{i}|x\right)$, where i∈1,2,…,C and *C* is the total fellow of the class, *j* depicts the index of the classifier, *n* is linked to the total number of classifiers, $a_{j}$ outlines the adaptive weight and *w* is the class tag of the result of the merger. In this fusion method, the multi-class relevance vector machines (RVM) [140] have been used as base classifiers.

#### 5.3.20. Activity and Device Position Recognition

Activity and device position recognition approach (ADPR) [141] merges the data from an accelerometer and multiple light sensors to classify the activities and positions of the devices. This method consists of two branches. The first branch calculates the state of motion and the position of the device using data from the accelerometer and a Bayesian classifier. The second branch refines the position estimates using the ambient light sensor, the proximity sensor and the intensity data of the camera, as well as the rules of whether one side of the device is occluded, if both sides are occluded or if none of the faces is occluded. The output of this second branch is a list of feasible device positions. The final classification of the movement state occurs by marginalizing the position of the device and vice-versa and eliminates the non-feasible positions. For reliability, a confidence metric is calculated and a decision of the classifier is made only when the confidence metric is above a threshold. Because this method combines the decision of the Bayesian classifier with the decision based on the rules of the second branch, this approach is classified in the merger at the decision level.

#### 5.3.21. Daily Activity Recognition Algorithm

Daily Activity Recognition Algorithm (DARA) [142] fuses the decisions of two classifiers to recognize human activities. This algorithm obtains the features (mean, variance and covariance) of the raw data from two inertial sensors. These features are introduced in two neural networks, one for each sensor. The outputs of two neural networks are fed into of a fusion module, which integrates these outputs (based on rules) and generates coarse-grained classification for three types of human activities—zero-displacement activities, transitional activities and strong displacement activities. Next, a heuristic discrimination module is used to accurately classify zero-displacement activities (such as sitting and standing) and transition activities (such as standing and standing to sit). Finally, a hidden Markov model (HMM)-based recognition algorithm is used for the detailed classification of strong displacement activities (for example, walking, climbing stairs, walking down stairs, running).

#### 5.3.22. Activity Recognition Model Based on Multibody-Worn Sensors

Activity Recognition Model Based on Multibody-Worn Sensors (ARMBMWS) [143] fuses the classification results based on Bayes’ theorem. In this model, each sensor node captures the raw activity data and extracts the features from sensor data stream. Then, the features of each sensor feed a decision tree classifier, one for each sensor. The final classification based on a Bayesian Naïve classifier is obtained by merging the result of each classifier. This Bayesian Naïve classifier classifies the entry instance to the class that maximizes the posterior probability.

#### 5.3.23. Physical Activity Recognition System

Physical activity recognition system (PARS) [144] fuses the decision of diverse classifiers. In this system, the temporal features and the Cepstral features of the raw data of the sensors are extracted. Temporary features are introduced in the Support Vector Machine (SVM) with the generalized linear discriminative sequence (GLDS) kernel and the Cepstral functions are introduced in the Gaussian Mixing Models (GMM) with the Heteroscedastic linear discriminant analysis (HLDA). The output of these models (SVM and GMM) are combined at the score level. This Score level fusion is defined as follows—Let us suppose that *K* classifiers exist and that each of them recognizes physical activities using a set of characteristics of a given sensor. Also, suppose that the *k*th classifier emits its own normalized logarithmic likelihood vector $l_{k}\left(x_{t}\right)$ for each test. Then, the logarithmic likelihood vector combined is defined by $\overset{´}{l}\left(x_{t}\right)=\sum_{k=1}^{K}\beta_{k}l_{k}\left(x_{t}\right)$. The weight, $\beta_{k}$, is obtained by logistic regression based on the training data [145].

#### 5.3.24. Distributed Activity Recognition through Consensus

Distributed Activity Recognition through Consensus (DARTC) [146] merges similarity scores from adjacent cameras to produce a probability for each action at the network level. In this method, each camera calculates a measure of similarity between the activities perceived by it and a dictionary of predefined activities. Also, it knows the likelihood of transition between activities. Based on these computed similarities and the probability of transitions, the consensus estimate is computed. The consensus is a likelihood of similarity of the activity seen against the dictionary, taking into consideration the decisions of the individual cameras. Basically, the consensus is the descending gradient algorithm. It minimizes the cost function $g\left(w_{i}\right)=\left(1/2\right)\sum_{j\in C_{i}^{n}}{\left(w_{i}-w_{j}\right)}^{2}$.

#### 5.3.25. A Hybrid Discriminative/Generative Approach for Modeling Human Activities

In the Discriminative/Generative Approach for Modeling Human Activities (DGAMHA) [147], a feature vector is calculated from raw data from the sensors. The vector includes linear and Mel-scale FFT frequency coefficients, cepstral coefficients, spectral entropy, bandpass filter coefficients, integrals, mean and variances. From this vector of characteristics, the fifty main characteristics per class are extracted and entered into an AdaBoost variation [148] that uses the decision stumps [149] as weak classifiers. Each decision stump classifier produces a verge in time *t*. This series of margins became probability when adjusted to a sigmoid. The distribution of probabilities is provided to ten Hidden Markov Models classifiers, each of which yields a probability. The most likely class is the classified class.

## 6. Comparison of Fusion Methods

In this section, first, we compared the fusion methods by the number of fusion methods that were combined, by the type of sensors (external and wearable) and by the extraction strategy of the features (manual and automatic). Then, we compare the scenarios addressed by the fusion methods to know which of them were the most used. After that, we compare the main elements (sensors, activities, classifiers and metrics) used by the fusion methods, to know which of them were the most used. Finally, we discuss both our findings on these comparisons and the limitations of this survey.

### 6.1. Comparison between Fusion Methods that Use a Single Fusion Method and Fusion Methods that Use Two Fusion Methods

In Table 5, we present the accuracies (minimum, average and maximum) reached as a whole by a group, which we call “Unmixed”. This group contains methods that were classified into a single fusion category (data-level, feature-level or decision-level). Also, in this Table, we show the accuracy (minimum, average and maximum) reached as a whole by a group, which we call “Mixed.” This group contains pairs of fusion methods that were used together. Each of these methods that form the pairs fits into only one of these categories.

In Table 5, for the calculation of the accuracies (minimum, average and maximum), we perform the next steps. (1) In the cases where there was a single paper that used one of the fusion methods or a couple of these methods, we took the highest accuracy of those reported in that paper as the representative performance of such a method or pair of methods. (2) In the cases where there were two or more papers that used the same fusion method or the same pair of these methods, we took the highest accuracy of those reported by each of those papers. Then, from these accuracies, we took the highest one as the representative performance of such a method or pair of methods. We take the maximum accuracy of fusion methods in all the above cases because we are interested in the maximum potential that these methods could achieve. 3) From these representative performances, one for each method or pair of methods, we calculate the minimum accuracy, the average accuracy and the maximum accuracy of the “Unmixed” group and the “Mixed” group.

In Table 5, we observe that the “Unmixed” group and the “Mixed” group got the same average accuracy (0.95). This result suggests that mixing two fusion methods is as competitive as using a single fusion method.

Also, in Table 5, we can see that the group “Mixed” shows a smaller range (0.044) than the range (0.336) of the “Unmixed” group. The range is a commonly used dispersion measure that calculates the difference between the maximum value and the minimum value in the data [150]. In our case, the value corresponds to the accuracy and the data correspond to some of these groups. We use the range because measures of central tendency (such as the mean) are not sufficient to describe the data (for example, two data sets can have the equal average but can be completely different); It is required to know its amplitude of variability [150]. This result suggests that methods that mix fusion methods are more consistent (in terms of accuracy) than methods that use a single method of fusion.

Besides, Table 5 shows that most of the proposed fusion methods (28/33) fit only one of the fusion categories (“Unmixed” group ). This result suggests that the mixture of fusion methods that are classified into different categories of fusion has been less explored than the use of a single fusion method.

Furthermore, in Table 5, it is possible to observe that most of the merging methods of the “Unmixed” group belong to the decision-level category (22/28). Also, we can see that most of the methods that belong to the “Mixed” group use a merging method that conforms to the decision-level category and a merging method that fits the feature-level category (4/5). These results suggest a trend towards the development of fusion methods that conform to the category of decision level.

### 6.2. Comparison between Fusion Methods that Merge Homogeneous Sensors and Fusion Methods That Combine Heterogeneous Sensors

In Table 6, we present the accuracies (minimum, average and maximum) reached as a whole by a group, which we call “Heterogeneous fused sensors.” This group contains both fusion methods and pairs of them that merge data from heterogeneous sensors (sensors of different types, such as accelerometers and gyroscopes). Also, in this Table, we show the accuracy (minimum, average and maximum) reached as a whole by a group, which we call “Homogeneous fused sensors.” This group contains both fusion methods and pairs of them that mix data from homogeneous sensors (sensors of the same type).

In Table 6, for the calculation of the accuracies (minimum, average and maximum), we perform the next steps. (1) In cases where there was a single work that used one of the fusion methods or a couple of these methods, we took the highest accuracy of those reported in that work as the representative performance of such a method or pair of methods. (2) In the cases where there were two or more articles that used the same fusion method or the same pair of these methods, we took the highest accuracy of those reported by each of those articles. Then, from these accuracies, we use the highest one as the representative performance of such a method or pair of methods. We take the maximum accuracy of the fusion methods in all the above cases as a measure of the maximum potential that these methods could achieve. (3) From these representative performances, one for each method or pair of methods, we calculate the minimum accuracy, the average accuracy and the maximum accuracy of the “Heterogeneous fused sensors” group and the “Homogeneous fused sensors” group.

In Table 6, we can see that the group of “Heterogeneous fused sensors” shows higher average accuracy than the group of “Homogeneous fused sensors.” This result suggests that the mixture of data from heterogeneous sensors produces more discriminative information than the mixture of homogeneous sensor data. Fusion methods could better exploit such information. Also, we can see that most of the proposed fusion methods or pairs of them (25/36) mix data from heterogeneous sensors. This result suggests a tendency to mix data from heterogeneous sensors, in the context of fusion methods.

### 6.3. Comparison between Fusion Methods That Automatically Extract Features and Fusion Methods That Manually Extract Features

In Table 7, we present the accuracies (minimum, average and maximum) reached as a whole by a group, which we call “Manual feature extraction.” This group contains fusion methods and pairs of them, which extract the features manually; such as extracting statistical features by hand—mean, standard deviation, to mention a few. Besides, in this Table, we show the accuracies (minimum, average and maximum) reached as a whole by a group, which we call “Automatic feature extraction.” This group contains fusion methods and pairs of them that extract the features automatically; for example, using CNN. Also, in this Table, we show the accuracies (minimum, average and maximum) reached as a whole by a group, which we call “Manual and automatic extraction of features.” This group contains fusion methods and pairs of them that extract features both manually and automatically.

In Table 7, for the calculation of the accuracies (minimum, average and maximum), we perform the next steps. (1) In cases where there was a single work that used one of the fusion methods or a couple of these methods, we took the highest precision of those reported in that work as the representative performance of such a method or pair of methods. (2) In cases where there were two or more articles that used the same fusion method or the same pair of these methods, we took the highest accuracy of those reported by each of those articles. Then, from these accuracies, we took the highest one as the representative performance of such a method or pair of methods. We take the maximum accuracy of fusion techniques in all the above cases because we are interested in the maximum potential that these approaches could reach. We take the maximum accuracy of fusion techniques in all the above cases because we are interested in the maximum potential that these approaches could reach. (3) From these representative performances, one for each method or pair of methods, we calculate the minimum accuracy, the average accuracy and the maximum accuracy of the group “Manual feature extraction”, the group “Automatic feature extraction” and the group “Manual and automatic extraction of features.”

In Table 7, we can see that the group “Manual feature extraction” shows a slightly higher average accuracy than the “Automatic feature extraction” group. Only one percentage point of difference between both groups. This result suggests that the automatic extraction of features is as competitive as the manual extraction of features.

Also, in Table 7, we can see that the group “Automatic feature extraction” shows a smaller difference between the maximum accuracy and the minimum accuracy (0.077 range) than the difference between the maximum accuracy and the minimum accuracy (0.336 range) of the “Manual feature extraction" group. This result suggests that fusion methods that automatically extract features are more consistent (in terms of accuracy) than fusion methods that manually extract features.

Besides, in Table 7, we can see that most of the proposed fusion methods (29/33) manually extract the features. This result indicates that fusion methods that automatically extract the characteristics have been less explored than methods that manually extract the features. This result also suggests that fusion methods that extract features manually and automatically have been less studied than methods that manually extract the characteristics.

Furthermore, in Table 7, we can see a promising accuracy in the fusion method that extracts the features manually and automatically (FA implemented by Ravi et al. [23]). This accuracy is higher than the average accuracy achieved in both the “Automatic feature Extraction” group and the “Manual feature Extraction” group.

### 6.4. Scenarios Most Used by Fusion Methods

In Table 8, we present scenarios addressed by fusion methods. These scenarios were identified by analyzing the types of activities that were recognized by the fusion methods. We found three types of scenarios. The first scenario is the “Activities of daily life,” which represents the activities that people usually perform to carry out their daily life, such as walking, running, jogging; to mention some. The second scenario is “Predetermined laboratory exercises,” which refers to sequences of activities designed by researchers, for example, walking to falling to lying, Walk right-circle; to name a few. The last scenario is “Situation in the medical environment,” which represents activities of some treatment or symptoms of a disease, for instance, actions of self-injection of insulin, hand flapping, and so forth.

In Table 8, we can see that the scenario most used by fusion methods is the “Activities of daily life.” 28 of 33 fusion methods address this scenario. This result suggests a tendency to recognize activities of the daily live by the fusion methods.

Also, in Table 8, we can see that the least used scenario for such methods is “Situation in the medical environment.” 3 of 33 fusion methods use this scenario. Besides, we note that the methods used in this scenario are based on ANNs.

Furthermore, in Table 8, we can see that the FA method addresses the three scenarios found and that the MulVS method uses two scenarios (“Activities of daily life” and “Predetermined laboratory exercise”).

### 6.5. Components Most Used by Fusion Methods

In Table 9, Table 10 and Table 11, we summarize the documents considered here (see Section 4). In these tables, we note that the most commonly used sensors are the accelerometers (54 times) and the gyroscope (32 times). None of the remaining sensors was used more than 18 times. This remark is consistent with what was reported by Jovanov et al. [18] and by Zhang et al. [19].

Besides, in Table 9, Table 10 and Table 11, we observed that the preferable activities to infer are “simple,” such as walking, running, climbing stairs, going downstairs, to name a few. The usual data sets used are benchmark data sets (such as WISDM v1.1 [151], Daphnet FoG [152], KARD [153], just to mention some), although the data set created on purpose is an evident practice (see Table 9, Table 10, Table 11 and Table 12).

In Table 9, Table 10 and Table 11, we also see that the most used classifiers are some ANN and SVM (27 times each). None of the remaining classifiers was used more than 12 times. Besides, we note that most of the classifiers used are not of the ANN type (12/13). Non-ANN classifiers are SVM, KNN, DT, NB, LR, RFC, Gaussian mixture models (GMM), Hidden Markov Model (HMM) [92], Conditional Random Field (CRF) [154], Multiclass relevance vector machines (RVM) [140], Bayesian networks (BN), Rule-based classifiers (RulBC), such as PART and NNGE [155], and Decision stump (Ds). This finding suggests that the ANN is a method that is gaining popularity in recognition of the activity. However, the cost-benefit balance with respect to the processing and accuracy of this ANN is unclear compared to non-ANN classifiers. Also, the main metrics used are the accuracy (64 times), the recall (22 times), the precision (18 times), and the F1-score (14 times).

Finally, after analyzing the papers considered here, we note that none of those papers explains the reason for choosing the fusion method they propose, nor the reasons why this fusion method works for a given data set.

### 6.6. Discussion and Trends

In this survey, we found that methods that combine two fusion methods that fit different fusion classifications achieved a performance (average accuracy) as good as methods using a unique fusion method. However, these methods that combine two fusion method were the most consistent (the “Mixed” group got the lowest range of 0.044) and less explored. These findings suggest that the combination of these methods is an emerging option, so knowing which of these methods could be combined optimally from a performance standpoint is a research gap that arises accordingly.

We also observed a tendency to develop methods that merge at the decision level. This finding suggests that the fusion at the decision level is an active field of investigation.

On the other hand, we noticed that fusion methods that combine heterogeneous sensors achieved better performance (in terms of average accuracy) than methods that combine homogeneous sensors. Also, we observed a tendency to develop fusion methods that mix heterogeneous sensors. This finding suggests that the fusion of heterogeneous sensors could be one of the first options when the performance is the target in applications based on HAR. Also, this finding opens a research gap to know what types of sensors could be combined optimally with performance in mind.

Besides, we found that the fusion methods that automatically extract the features achieved an average accuracy as good as the fusion methods that manually extract the characteristics. However, the fusion methods that automatically extract the features were the most consistent (the group “Automatic feature extraction” obtained the lowest range of 0.077) and less explored. These results suggest that fusion methods that include automatic feature extraction are an emerging option, in the context of HAR. Also, these findings suggest a gap in research to know the optimal model of deep learning, in terms of accuracy and time, to automatically extract characteristics and recognize human activities.

Also, we located an FA implementation [23] that extracts characteristics manually and automatically with a promising performance (see Table 7). This suggests more research is required to explore the potential of combining automatic feature extraction and manual feature extraction.

Moreover, we noticed a tendency to recognize the activities of daily life through fusion methods. This result suggests that the recognition of activities of daily living by fusion methods is an active field of research. One reason that could motivate this trend is that not all data fusion methods are adequate in all cases (data sets) [175].

We also learned that the “Situation in the medical environment” is the scenario least addressed by the fusion methods and that the fusion methods that use this scenario are based on the ANNs. These results suggest that the recognition of activities in the context of the medical scenario through the use of ANNs is a coming up area, so knowing the appropriate model of these ANNs for these activities is an emerging research gap.

Likewise, we located only two fusion methods (FA and MulVS) that address at least two scenarios, so exploring the behavior of the rest of these methods in the the three scenarios is a research gap that arises accordingly.

Furthermore, we found that the papers studied here do not explain the reason for choosing the fusion method they propose, nor the reasons why this fusion method works for a given data set. This finding suggests that researchers may have trouble choosing some method of fusion for a particular data set. When they want to combine information from various sources, they resort to trial and error or, even worse, they use the fusion methods they know [175]. To address this problem of choosing a fusion method, Aguileta et al. [175] proposed a method to predict the optimal fusion method for a given data set that stores human activities. However, although this method is promising (it predicts with an accuracy of 0.9), it only considers eight fusion methods and 65 original data sets with human activity data collected by accelerometers and gyroscopes. More cross-sectional studies are needed between different combinations of data sets, classifiers and fusion methods that guide us to choose the best algorithms and their combinations to infer human activities according to the characteristics of a particular data set.

### 6.7. Study Limitations

This survey was based on a systematic mapping approach [176]. However, secondary works such as the one reported here are subject to restrictions. The typical restrictions that can occur in a mapping study are data extraction error (limited coverage), the selection of academic search engines and the researcher’s bias during the mapping study process, such as selection of articles, recovery of data, analysis, and synthesis. Now we explain how these restrictions were approached.

The restriction of the selected search terms and search engines can lead to an inadequate set of primary studies. We addressed this problem by selecting the Scopus database, which involves a broad spectrum of peer-reviewed articles and a user-friendly interface for advanced search capabilities.

To make this survey repeatable for other researchers, the search engine, the search terms and the inclusion/exclusion criteria were strictly defined and informed. However, it is necessary to bear in mind that the search terms we use are related to the recognition of human activity based on the fusion of data from multiple sensors; existing relevant papers that do not contain any of the terms used may have been missed. However, the relevant documents identified are a representative sample that serves to make a drawing on the subject and provide a generalization of the current state of the fusion methods used in recognition of human activity.

Our findings are based on articles published in English, and papers published in languages other than English were excluded from this study. We consider that the grouped documents contain enough information to represent the informed knowledge on the subject.

The application of the inclusion and exclusion criteria and the categorization of the documents may be affected by the judgment and experience of the investigators, and there could have been a personal bias. To lessen this bias, joint voting was utilized in the selection and categorization of the document; differences were solved by consensus among the authors of this document.

## 7. Conclusions

Multisensor fusion, in the context of HAR, is an active research field that is growing significantly, and there is such a variety of methods that it is often difficult to choose some of them for a particular situation. So, organizing these methods is an action that seems obvious. In the literature, some works examine and classify fusion methods under some classification but these works mainly limit the type of sensors to be studied and address specific aspects of the fusion process.

In this paper, we have presented a survey of the state of the art of the literature on contributions to the fusion of multi-sensor (external and wearable) data, in the context of HAR. We have based this survey on a systematic mapping approach to find relevant works.

We had made a big effort to organize the many different works into the main families of fusion methods for HAR (data level, feature level, and decision level, as suggested by Liggins [34]), and we have organized them in variations and combinations of the main categories, task that is extremely hard given the big amount of combinations of methods of different nature in a single system that is often found. We have thus developed a systematic and organized comparison of the different works, which is the main substance of this survey.

After analyzing these articles, we have identified and compared the performance of methods that use a single fusion method and methods that use two fusion methods. Also, we have examined the performance of techniques that merge homogeneous sensors and approaches that combine heterogeneous sensors. Similarly, we have identified and compared approaches that manually extract features and methods that automatically extract characteristics. Further, we have identified the scenarios most used by the fusion methods and some of the components most used by these methods, such as sensors, activities, classifiers, and metrics. Finally, we have discussed relevant directions and future challenges on fusion methods in the HAR context, as well as the limitations of this work.

## Figures and Tables

**Figure 1 sensors-19-03808-f001:**
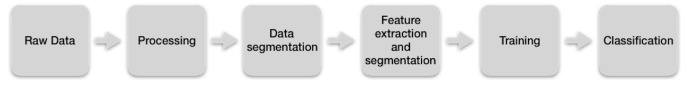
Activity recognition workflow.

**Figure 2 sensors-19-03808-f002:**
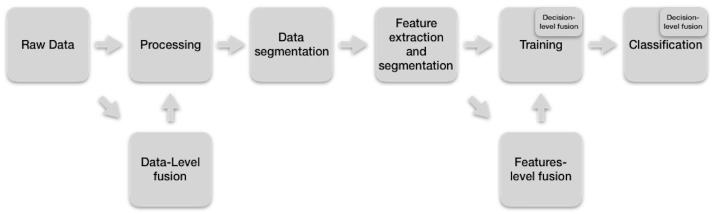
Extended activity recognition workflow.

**Figure 3 sensors-19-03808-f003:**
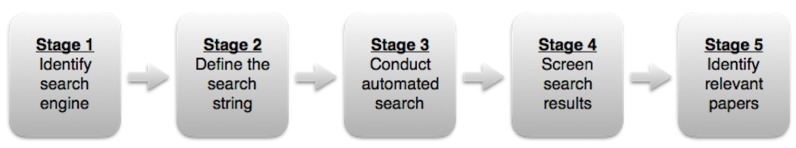
Mapping study stages.

**Table 1 sensors-19-03808-t001:** Previous surveys comparison, including ours.

Criterion	Gravina [20]	Chen [30]	Shivappa [31]	Ours
Classification at the data level	Yes	Yes	No	Yes
Classification at the feature level	Yes	Yes	Yes	Yes
Classification at the decision level	Yes	Yes	Yes	Yes
Classification signal enhancement and sensor level fusion strategies, classification at the classifier level, classification facilitating the natural interaction of the human-computer and classification exploitation of complementary information through modalities	No	No	Yes	No
Mixed fusion	No	No	Yes	Yes
Physical activity recognition	Yes	Yes	No	Yes
Emotion recognition	Yes	No	Yes	No
Speech recognition	No	No	Yes	No
Tracking	No	No	Yes	Yes
Biometrics	No	No	Yes	No
Meeting scene analysis	No	No	Yes	No
Fusion in the context of general health	Yes	No	No	Yes
Fusion characteristics	Yes	No	No	No
Fusion parameters	Yes	Yes	Yes	No
Type of sensors	Wearable	Depth cameras and inertial sensors	Microphones and cameras	External and wearable
Activities	Yes	Yes	No	Yes
Datasets	No	Yes	Yes	Yes
Classifiers	Yes	Yes	Yes	Yes
Metrics	No	Yes	Yes	Yes
Explanation of fusion methods	General	General	General	Detailed
Homogeneous sensors vs heterogeneous sensors	No	No	No	Yes
Automatic feature extraction vs Manual feature extraction	No	No	No	Yes
Unmixed Fusion vs Mixed fusion	No	No	No	Yes

**Table 2 sensors-19-03808-t002:** Search string defined.

Search String
(multi or diverse) AND sensor AND data AND (fusion OR combine) AND human AND (activity OR activities) AND (recognition OR discover OR recognize)

**Table 3 sensors-19-03808-t003:** Inclusion and exclusion criteria.

Criteria	Description
IC1	Include papers whose titles are related to the recognition of human activities through multiple modalities
IC2	Include papers that contain terms related with the defined terms in the search string.
IC3	Include papers whose abstracts are related to the recognition of human activities through multiple sensors
IC4	Include publications that have been partially or fully read.
EC1	Exclude documents written in languages other than English

**Table 4 sensors-19-03808-t004:** Document search results and relevant selected papers.

Source	Document Results	Relevant Papers
Scopus	78	33

**Table 5 sensors-19-03808-t005:** The minimum, average and maximum accuracy reached by the “Unmixed” group and by the “Mixed” group.

Unmixed Group	Acurracy	Mixed Group	Accuracy
**Data-level fusion**			
(1) TLSF			
**Feature-level fusion**			
(2) FA, (3) FA-PCA, (4) TF, (5) DRP and JSR, (6) SVMBMF			
**Decision-level fusion**			
(7) LBEL, (8) SMMKL, (9) a-stack, (10) Vot, (11) AdaBoost, (12) MulVS, (13) HWC, (14) Prod, (15) Sum, (16) Max, (17) Min, (18) Ran, (19) WA, (20) CMMSDF, (21) MFN, (22) GABCEO, (23) WLOGP, (24) ADPR, (25) DARA, (26) ARMBMWS, (27) PARS and (28) DARTC	Minimum: 0.664 Average: 0.95 Maximum: 1	(1) TF-RDA, (2) UbiMonitorIF-FA, (3) GABCF-FC, (4) MBSSAHC-FA, and (5) DGAMHA-FA	Minimum: 0.927 Average: 0.95 Maximum: 0.971

**Table 6 sensors-19-03808-t006:** The minimum, average and maximum accuracy reached by the “Homogeneous fused sensors” group and by the “Heterogeneous fused sensors” group.

Homogeneous Fused Sensors Group	Accuracy	Heterogeneous Fused Sensors Group	Accuracy
**Feature-level fusion**	Minimum: 0.664 Average: 0.923 Maximum: 1	**Data-level fusion**	Minimum: 0.881 Average: 0.962 Maximum: 1
(1) FA		(1) TLSF	
		**Feature-level fusion**	
		(2) FA, (3) FA-PCA, (4) TF, (5) DRP and JSR, and (6) SVMBMF	
**Decision-level Fusion**		**Decision-level Fusion**	
(2) SMMKL, (3) Vot, (4) HWC, (5) CMMSDF, (6) ADPR, (7) ARMBMWS, and (8) DARTC		(7) LBEL, (8) A-stack, (9) Vot, (10) AdaBoost, (11) MulVS, (12) Prod, (13) Sum, (14) Max, (15) Min, (16) Ran, (17) WA, (18) MFN, (19) GABCEO, (20) WLOGP, (21) DARA, and (22) PARS	
**Two-level Fusion**		**Two-level Fusion**	
(9) TF-RDA, (10) MBSSAHC-FA and (11) UbiMonitorIF-FA		(23) TF-RDA, (24) GABCF-FC and (25) DGAMHA-FA	

**Table 7 sensors-19-03808-t007:** The minimum, average and maximum accuracy reached by the group “Manual feature extraction,” by the group “Automatic feature extraction,” and by the group “Manual and automatic extraction of features”.

Manual Feature Extraction	Accuracy	Automatic Feature Extraction	Accuracy	Manual and Automatic Extraction of Features	Accuracy
**Data-Level Fusion**		**Feature-Level Fusion**		**Feature-Level Fusion**	
(1) TLSF		(1) TF			
**Feature-level Fusion**		**Decision-level Fusion**			
(2) FA, (3) FA-PCA, (4) DRP and JSR, (5) SVMBMF		(2) a-stack			
**Decision-level Fusion**		**Two-level fusion**			
(6) LBEL, (7) SMMKL, (8) Vot, (9) AdaBoost, (10) MulVS, (11) HWC, (12) Prod, (13) Sum, (14) Max, (15) Min, (16) Ran, (17) WA, (18) CMMSDF, (19) MFN, (20) GABCEO, (21) WLOGP, (22) ADPR, (23) DARA, (24) ARMBMWS, and (25) PARS	Minimum: 0.664 Average: 0.95 Maximum: 1	(3) TF-RDA	Minimum: 0.923 Average: 0.96 Maximum: 1	(1) FA	0.986
**Two-level fusion**					
(26) MBSSAHC-FA, (27) ubiMonitorIF-FA, (28) GABCF-FC and (29) DGAMHA-FA					

**Table 8 sensors-19-03808-t008:** Scenarios by fusion methods.

Fusion Method	Activities of Daily Life	Predetermined Laboratory Exercises	Situation in the Medical Environment
Data-level fusion			
TLSF		Yes	
Feature-level fusion			
FA	Yes	Yes	Yes
FA-PCA	Yes		
TF			Yes
DRP and JSR		Yes	
SVMBMF	Yes		
Decision-level fusion			
LBEL	Yes		
SMMKL		Yes	
a-stack	Yes		
Vot	Yes		
AdaBoost	Yes		
MulVS	Yes	Yes	
HWC	Yes		
Prod	Yes		
Sum	Yes		
Max	Yes		
Min	Yes		
Ran	Yes		
WA	Yes		
CMMSDF	Yes		
MFN	Yes		
GABCEO	Yes		
WLOGP		Yes	
ADPR	Yes		
DARA		Yes	
ARMBMWS	Yes		
PARS	Yes		
DARTC	Yes		
Two-level fusion			
TF-RDA			Yes
MBSSAHC-FA	Yes		
UbiMonitorIF-FA	Yes		
GABCF-FC	Yes		
DGAMHA-FA	Yes		

**Table 9 sensors-19-03808-t009:** Summary of articles that propose methods that combine data at the level of data or at the level of features. Ref = Reference and DId = Dataset ID. Acc(s) = Acceleromter(s), Mag(s) = Magnetometer(s), Gyr(s) = Gyroscope(s), Kc = Kinect camera, Hr = Heart rate, Av = Anthropometric variables, Res = Respiration, Press = Pressure Mic(s)= Microphone(s), Vid(s) = Videos(s), IMU = Inertial measurement unit, Mot = Motion, and Ven = Ventilation.

Ref	Fusion Method	Sensors	Activities	DId	Classifiers	Metrics
**Data-level fusion**						
[106]	TLSF	WiFi, and Acc	(1) following relations and (2) group leadership	1	SVM	Error %: 7
**Feature-level fusion**						
[108]	FA	Accs	(1) walking to falling to lying, and (2) sitting to falling to sitting down	2	SVM	Accuracy: 0.950 Detection rate: 0.827 False alarm rate: 0.05
[115]	FA	Accs	(1) walking, (2) upstairs, (3) downstairs, (4) sitting, (5) standing, and (6) lLying	3	KNN	Recall: 0.624 Precision: 0.941
[115]	FA	Gyrs	(1) walking, (2) upstairs, (3) downstairs, (4) sitting, (5) standing, and (6) lying down	3	KNN	Recall: 0.464 Precision: 0.852
[22]	FA	Acc, and Gyr	(1) walking, (2) sitting, (3) standing, (4) jogging, (5) biking, (6) walking upstairs, and (7) walking downstairs	4	BN	Accuracy: >0.6 and <1
[22]	FA	Acc, and Gyr	(1) walking, (2) sitting, (3) standing, (4) jogging, (5) biking, (6) walking upstairs, and (7) walking downstairs	4	NB	Accuracy: >0.4 and <1
[22]	FA	Acc, and Gyr	(1) walking, (2) sitting, (3) standing, (4) jogging, (5) biking, (6) walking upstairs, and (7) walking downstairs	4	LR	Accuracy: >0.6 and <1
[22]	FA	Acc, and Gyr	(1) walking, (2) sitting, (3) standing, (4) jogging, (5) biking, (6) walking upstairs, and (7) walking downstairs	4	SVM	Accuracy: >0.6 and <1
[22]	FA	Acc, and Gyr	(1) walking, (2) sitting, (3) standing, (4) jogging, (5) biking, (6) walking upstairs, and (7) walking downstairs	4	KNN	Accuracy: >0.8 and <1
[22]	FA	Acc, and Gyr	(1) walking, (2) sitting, (3) standing, (4) jogging, (5) biking, (6) walking upstairs, and (7) walking downstairs	4	DT	Accuracy: >0.8 and <1
[22]	FA	Acc, and Gyr	(1) walking, (2) sitting, (3) standing, (4) jogging, (5) biking, (6) walking upstairs, and (7) walking downstairs	4	RFC	Accuracy: >0.8 and <1
[22]	FA	Acc, and Gyr	(1) walking, (2) sitting, (3) standing, (4) jogging, (5) biking, (6) walking upstairs, and (7) walking downstairs	4	RulBC	Accuracy: >0.8 and <1
[23]	FA	Acc	(1) walk,(2) jog, (3) ascend stairs, (4) descend stairs, (5) sit, and (6) stand	5	CNN-ANN-SoftMax	Accuracy: 0.986 Precision:0.975 Recall:0.976
[23]	FA	Acc, and Gyr	(1) casual movement, (2) cycling, (3) no acivity (Idle), (4) public transport, (5) running, (6) standing and (7) walking	6	CNN-ANN-SoftMax	Accuracy: 0.957 Precision: 0.930 Recall: 0.933
[23]	FA	Acc	(1) walking, (2) jogging, (3) stairs, (4) sitting, (5) standing, and (6) lying Down	7	CNN-ANN-SoftMax	Accuracy: 0.927 Precision: 0.897 Recall: 0.882
[23]	FA	Acc	(1) write on notepad, (2) open hood, (3) close hood, (4) check gaps on the front door, (5) open left front door, (6) close left front door, (7) close both left door, (8) check trunk gaps, (9) open and close trunk, and (10) check steering	8	CNN-ANN-SoftMax	Accuracy: 0.953 Precision: 0.949 Recall: 0.946
[23]	FA	Acc	freezing of gait (FOG) symptom	9	CNN-ANN-SoftMax	Accuracy: 0.958 Precision: 0.826 Recall: 0.790
[109]	FA	Kc	(1) catch cap, (2) toss paper, (3) take umbrella, (4) walk, (5) phone call, (6) drink, (7) sit down, and (8) stand	10	SVM	Accuracy: 1
[109]	FA	Kc	(1) wave, (2) drink from a bottle, (3) answer phone, (4) clap, (5) tight lace, (6) sit down, (7) stand up, (8) read watch, and (9) bow	11	SVM	Accuracy: 0.904
[112]	FA	Acc, Hr, and Av	(1) Lying: Lying down resting; (2) low whole body motion (LWBM): Sitting resting, sitting stretching, standing stretching, desk work, reading, writing, working on a PC, watching TV, sitting fidgeting legs, standing still, bicep curls, shoulder press; (3) high whole body motion (HWBM): Stacking groceries, washing dishes, preparing a salad, folding clothes, cleaning and scrubbing, washing windows, sweeping, vacuuming; (4) Walking; (5) Biking; and (6) Running	12	DT	Accuracy: 0.929 Sensitivity:0.943 Specificity:0.980
[113]	FA	Accs, and Res	(1) Computer work, (2) Filing papers, (3) Vacuuming, (4) Moving the box, (5) Self-paced walk, (6) Cycling 300 kpm, (7) Cycling 600 kpm, (8) Level treadmill walking (3 mph), (9) Treadmill walking (3 mph and 5% grade), (10) Level treadmill waking (4 mph), (11) Treadmill walking (4 mph and 5% grade), (12) Level treadmill running(6 mph), (13) Singles tennis against a practice wall, and (14) Basketball	13	SVM	Accuracy: 0.79
[114]	FA	Mics and Vids	(1) eating-drinking, (2) reading, (3) ironing, (4) cleaning, (5) phone answering, and (6) TV watching	14	GMM	Accuracy: 0.6597
[110]	FA-PCA	Acc, Mag, Gyr, and Press	(1) sitting, (2) standing, (3) walking, (4) running, (5) cycling, (6) stair descent, (7) stair ascent, (8) elevator descent, and (9) elevator ascent	15	DT	Accuracy: 0.894
[110]	FA-PCA	Acc, Mag, Gyr, and Press	(1) sitting, (2) standing, (3) walking, (4) running, (5) cycling, (6) stair descent, (7) stair ascent, (8) elevator descent, and (9) elevator ascent	15	MLP	Accuracy: 0.928
[110]	FA-PCA	Acc, Mag, Gyr, and Press	(1) sitting, (2) standing, (3) walking, (4) running, (5) cycling, (6) stair descent, (7) stair ascent, (8) elevator descent, and 9) elevator ascent	15	SVM	Accuracy: 0.928
[110]	FA-PCA	Acc, Mag, Gyr, and Press	(1) sitting, (2) standing, (3) walking, (4) running, (5) cycling, (6) stair descent, (7) stair ascent, (8) elevator descent, and (9) elevator ascent	15	NB	Accuracy: 0.872
[111]	FA-PCA	IMU and Press	(1) sitting, (2) standing, and (3) walking	16	SVM	Accuracy: 0.99
[48]	TF	Vid and Mot	activity of self-injection of insulin includes 7 action class: (1) Sanitize hand, (2) Roll insulin bottle (3) Pull air into syringe, (4) Withdraw insulin, (5) Clean injection site, (6) Inject insulin, and (7) Dispose needle	17	CNN-LSTM-Softmax	Accuracy: 1
[119]	DRP and JSR	Acc and Gyr	(1) Stand, (2) Sit, (3) Lie down, (4) Walk forward, (5) Walk left-circle, (6) Walk right-circle, (7) Turn left, (8) Turn right, (9) Go upstairs, (10) Go downstairs, (11) Jog, (12) Jump, and (13) Push wheelchair	18		Accuracy: 0.887
[120]	SVM BMF	Acc and Ven	(1) Computer work, (2) filing papers, (3) vacuuming, (4) moving boxes, (5) self-paced walk, (6) cycling, (7) treadmill, (8) backed ball, and (10) tennis	19	SVM	Accuracy: 0.881

**Table 10 sensors-19-03808-t010:** Summary of works proposing methods that combine data at the decision level. Ref = Reference and DId = Dataset ID. Acc(s) = Acceleromter(s), Mag(s) = Magnetometer(s), Gyr(s) = Gyroscope(s), ECG = Electrocardiography, Hr = Heart rate, Alt = Altimeter, Tem = Temperature, Bar = Barometer, Lig = Light, Mot = Motion, ElU = Electricity usage, Mic(s) = Microphone(s), OMCS = Optical motion capture system, Kc = Kinect camera, and Vid(s) = Videos(s).

Ref	Fusion Method	Sensors	Activities	DId	Classifiers	Metrics
**Decision-level fusion**						
[123]	LBEL	Acc, and iBeacon [156]	(1) standing, (2) walking, (3) cycling, (4) lying, (5) sitting, (6) exercising, (7) prepare food, (8) dining, (9) watching TV, (10) prepare clothes, (11) studying, (12) sleeping, (13) bathrooming, (14) cooking, (15) past times, and (16) random	20	DT	Accuracy: 0.945
[108]	SMM KL	Accs	(1) walking to falling to lying, and (2) sitting to falling to sitting down	3	MKL-SVM	Accuracy: 0.946 Detection rate: 0.347 False alarm rate: 0.05
[157]	a-stack	Acc, Gyr, ECG, and Mag	(1) lying, (2) sitting/standing, (3) walking, (4) running, (5) cycling, and (6) other	21	NN, LR	Accuracy: 0.923
[157]	a-stack	Acc, Hr, Gyr, and Mag	(1) lying, (2) sitting/standing, (3) walking, (4) running, (5) cycling, and (6) other	22	NN, LR	Accuracy: 0.848
[158]	Vot	Acc	(1) Walking, (2) Jogging, (3) Upstairs, (4) Downstairs, (5) Sitting, and (6) Standing	5	MLP, LR, and DT	Accuracy: 0.916 AUC: 0.993 F1-score: 0.918
[138]	Vot	Acc, Alt, Tem, Gyr, Bar, lig, and Hr	(1) brushing teeth, (2) exercising, (3) feeding, (4) ironing, (5) reading (6) scrubbing, (7) sleeping, (8) using stairs, (9) sweeping, (10) walking, (11) washing dishes, (12) watching TV, and (13) wiping	22	MLP, RBF, and SVM	Accuracy: 0.971
[137]	Vot	Mot, and Tem	(1) Wash Dishes, (2) Watch TV, (3) Enter Home, (4) Leave Home, (5) Cook Breakfast, (6) Cook Lunch, (7) Group Meeting, and (8) Eat Breakfast	24	ANN, HMM, CRF, SVM	Accuracy: 0.906 Precision: 0.799 Recall: 0.7971 F1-score: 0.7984
[137]	Vot	Mot, Door, and Tem	(1) bed to toilet, (2) sleeping, (3) leave home, (4) watch TV, (5) chores, (6) desk activity, (7) dining, (8) evening medicines, (9) guest bathroom, (10) kitchen activity, (11) master bathroom, (12) Master Bedroom, (13) meditate, (14) morning medicines, and (15) read	25	ANN, HMM, CRF, SVM	Accuracy: 0.885 Precision: 0.801 Recall: 0.8478 F1-score: 0.8235
[137]	Vot	Mot, item, Door, tem, ElU, and Lig	(1) meal preparation, (2) sleeping, (3) cleaning, (4) work, (5) grooming, (6) shower, and (7) wakeup	26	ANN, HMM, CRF, SVM	Accuracy: 0.855 Precision: 0.752 Recall: 0.7274 F1-score: 0.7394
[137]	Ada Boost	Mot and Tem	(1) wash dishes, (2) watch TV, (3) enter home, (4) leave home, (5) cook breakfast, (6) cook lunch, (7) group meeting, and (8) eat breakfast	24	DT	Accuracy: 0.912 Precision: 0.844 Recall: 0.7983 F1-score: 0.8206
[137]	Ada Boost	Mot, Door, and Tem	(1) bed to toilet, (2) sleeping, (3) leave home, (4) watch TV, (5) chores, (6) desk activity, (7) dining, (8) evening medicines, (9) guest bathroom, (10) kitchen activity, (11) master bathroom, (12) master bedroom, (13) meditate, (14) morning medicines, and (15) read	25	DT	Accuracy: 0.875 Precision: 0.824 Recall: 0.8767 F1-score: 0.805
[137]	Ada Boost	Mot, item, Door, Tem, ElU, and Lig	(1) meal preparation, (2) sleeping, (3) cleaning, (4) work, (5) grooming, (6) shower, and (7) qakeup	26	DT	Accuracy: 0.837 Precision: 0.736 Recall: 0.7174 F1-score: 0.7266
[27]	MulVS	Mic, and Acc	(1) mop floor, (2) sweep floor, (3) type on computer keyboard, (4) brush teeth, (5) wash hands, (6) eat chips, and (7) watch TV	27	RFC	Accuracy: 0.941 Recall: 0.939 Specificity: 0.99
[27]	MulVS	Mic, Acc, and OMCS	(1) jumping in place, (2) jumping jacks, (3) bending, (4) punching, (5) waving two hands, (6) waving one hand, (7) clapping, (8) throwing a ball, (9) sit/stand up, (10) sit down, and (11) stand up	28	RFC	Accuracy: 0.995 Recall: 0.995 Specificity: 0.99
[27]	MulVS	Acc, Gyr, and Kc	(1) swipe left, (2) swipe right, (3) wave, (4) clap, (5) throw, (6) arm cross, (7) basketball shoot, (8) draw x, (9) draw circle CW, (10) draw circle CCW, (11) draw triangle, (12) bowling, (13) boxing, (14) baseball swing, 15) tennis swing, (16) arm curl, (17) tennis serve, (18) push, (19) knock, (20) catch, (21) pickup throw, (22) jog, (23) walk, (24) sit 2 stand, (25) stand 2 sit, (26) lunge, and (27) squat	29	RFC	Accuracy: 0.981 Recall: 0.984 Specificity: 0.99
[27]	MulVS	Acc, Gyr, and Mag	(1) stand, (2) walk, (3) sit, and (4) lie	30	RFC	Accuracy: 0.925 Recall: 0.905 Specificity: 0.96
[134]	HWC	Accs	(1) running, (2) cycling, (3) stretching, (4) strength-training, (5) walking, (6) climbing stairs, (7) sitting, (8) standing and (9) lying down	31	KNN	Accuracy: 0.975
[138]	Prod	Acc, Alt, Tem, Gyr, Bar, Lig, and Hr	(1) brushing teeth, (2) exercising, (3) feeding, (4) ironing, (5) reading, (6) scrubbing, (7) sleeping, (8) usingstairs, (9) sweeping, (10) walking, (11) washing dishes, (12) watching TV, and (13) wiping	23	MLP, RBF, and SVM	Accuracy: 0.972
[138]	Sum	Acc, Alt, Tem, Gyr, Bar, Lig, and Hr	(1) brushing teeth, (2) exercising, (3) feeding, (4) ironing, (5) reading, (6) scrubbing, (7) sleeping, (8) usingstairs, (9) sweeping, (10) walking, (11) washing dishes, (12) watching TV, and (13) wiping	23	MLP, RBF, and SVM	Accuracy: 0.973
[138]	Max	Acc, Alt, Tem, Gyr, Bar, Lig, and Hr	(1) brushing teeth, (2) exercising, (3) feeding, (4) ironing, (5) reading, (6) scrubbing, (7) sleeping, (8) usingstairs, (9) sweeping, (10) walking, (11) washing dishes, (12) watching TV, and (13) wiping	23	MLP, RBF, and SVM	Accuracy: 0.971
[138]	Min	Acc, Alt, Tem, Gyr, Bar, Lig, and Hr	(1) brushing teeth, (2) exercising, (3) feeding, (4) ironing, (5) reading, (6) scrubbing, (7) sleeping, (8) usingstairs, (9) sweeping, (10) walking, (11) washing dishes, (12) watching TV, and (13) wiping	23	MLP, RBF, and SVM	Accuracy: 0.971
[138]	Ran	Acc, Alt, Tem, Gyr, Bar, Lig, and Hr	(1) brushing teeth, (2) exercising, (3) feeding, (4) ironing, (5) reading, (6) scrubbing, (7) sleeping, (8) usingstairs, (9) sweeping, (10) walking, (11) washing dishes, (12) watching TV, and (13) wiping	23	MLP, RBF, and SVM	Accuracy: 0.969
[138]	WA	Acc, Alt, Tem, Gyr, Bar, Lig, and Hr	(1) brushing teeth, (2) exercising, (3) feeding, (4) ironing, (5) reading, (6) scrubbing, (7) sleeping, (8) using stairs, (9) sweeping, (10) walking, (11) washing dishes, (12) watching TV, and (13) wiping	23	MLP, RBF, and SVM	Accuracy: 0.971
[135]	CMM SDF	Mot	(1) using laptop, (2) watching TV, (3) eating, turning on the stove, and (5) washing dishes	32		Accuracy: 1
[136]	MFN	Kc, and Mic	Recognition of objects through human actions	33	SVM, and NB	Accuracy: 0.928 F1-score: 0.921
[137]	GAB CEO	Mot, and Tem	(1) wash dishes, (2) watch TV, (3) enter home, (4) leave home, (5) cook breakfast, (6) cook lunch, (7) group meeting, and (8) eat breakfast	24	ANN, HMM, CRF, SVM	Accuracy: 0.951 Precision: 0.897 Recall: 0.9058 F1-score: 0.9013
[137]	GAB CEO	Mot, door, and Tem	(1) bed to toilet, (2) sleeping, (3) leave home, (4) watch TV, (5) chores, (6) desk activity, (7) dining, (8) evening medicines, (9) guest bathroom, (10) kitchen activity, (11) master bathroom, (12) master bedroom, (13) meditate, (14) morning medicines, and (15) read	25	ANN, HMM, CRF, SVM	Accuracy: 0.919 Precision: 0.827 Recall: 0.8903 F1-score: 0.8573
[137]	GAB CEO	Mot, item, Door, Tem, ElU, and lig	(1) meal preparation, (2) sleeping, (3) cleaning, (4) work, (5) grooming, (6) shower, and (7) wakeup	26	ANN, HMM, CRF, SVM	Accuracy: 0.894 Precision: 0.829 Recall: 0.8102 F1-score: 0.8197
[139]	WLO GP	Acc, and Gyr	(1) stand, (2) sit, (3) lie down, (4) walk forward, (5) walk left-circle, (6) walk right-circle, (7) turn left, (8) turn right, (9) go upstairs, (10) go downstairs, (11) jog, (12) jump, (13) push wheelchair	18	RVM	Accuracy: 0.9878
[141]	ADPR	Accs	(1) walk, (2) run, (3) sit, (4) stand, (5) fiddle, and (6) rest	34	NB, and GMM	F1-score: 0.926
[142]	DARA	Acc, and Gyr	(1) zero-displacement activities AZ = {standing, sitting, lying}; (2) transitional activities AT = {sitting-to-standing, standing-to- sitting, level walking-to-stair walking, stair walking-to-level walking, lying-to-sitting, sitting-to- lying}; and (3) strong dis- placement activities AS = {walking level, walking upstairs, walking downstairs, running}	35	ANN, and HMM	Accuracy: 0.983
[143]	ARM BMWS	Accs	(1) walking, (2) walking while carrying items, (3) sitting and relaxing, (4) working on computer, (5) standing still, (6) eating or drinking, (7) watching TV, (8) reading, (9) running, (10) bicycling, (11) stretching, (12) strength-training, (13) scrubbing, (14) vacuuming, (15) folding laundry, (16) lying down and relaxing, (17) brushing teeth, (18) climbing stairs, (19) riding elevator, and (20) riding escalator	31	NB, and DT	Accuracy: 0.6641
[144]	PARS	ECG, and Acc	(1) lying, (2) sitting, (3) sitting fidgeting, (4) standing, (5) standing fidgeting, (6) playing Nintendo Wii tennis, (7) slow walking, 8) brisk walking, and 9) running	36	SVM, and GMM	Accuracy: 0.973
[146]	DAR TC	Vids	(1) looking at watch, (2) scratching head, (3) sit, (4) wave hand, (5) punch, (6) kick, and (7) pointing a gun	37	Bayes rule and Markov chain.	Average probability of correct match: Between 3-1

**Table 11 sensors-19-03808-t011:** Summary of papers that propose methods that merge data at two levels. Ref = Reference and DId = Dataset ID. Acc(s) = Acceleromter(s), Mag(s) = Magnetometer(s), Gyr(s)= Gyroscope(s), Alt = Altimeter, Tem = Temperature, Bar = Barometer, Lig = Light, Hr= Heart rate, Mic(s) = Microphone(s), Hum = Humidity, and Com = Compass.

Ref	Fusion Method	Sensors	Activities	DId	Classifiers	Metrics
**Two-level fusion**						
[50]	TF-RDA	Acc, Gyr, and Mag	(1) hand flapping	38	CNN-LSTM-Softmax	F1-score: 0.95
[50]	TF-RDA	Accs	(1) body rocking, (2) hand flapping or (3) simultaneous body rocking, and hand flapping	39	CNN-LSTM-Softmax	F1-score: 0.75
[29]	MBS SAHC -FA	Accs	(1) lying, (2) sitting, (3) standing, (4) walking, (5) stairs, (6) transition	40	DT, and NB	Accuracy: 0.927
[133]	Ubi Moni torIF-FA	Accs	(1) lying, (2) sitting, (3) standing, (4) walking, (5) running, (6) cycling, (7) Nordic walking, (8) ascending stairs, (9) descending stairs, (10) vacuum cleaning, (11) ironing, and (12) rope jumping	22	DT, and SMV	Accuracy: 0.95 Precision: 0.937 Recall: 0.929 F1-score:0.93
[138]	GABC F-FC	Acc, Alt, Tem, Gyr, Bar, lig, and Hr	(1) brushing teeth, (2) exercising, (3) feeding, (4) ironing, (5) reading, (6) scrubbing, (7) sleeping, (8) using stairs, (9) sweeping, (10) walking, (11) washing dishes, (12) watching TV, and (13) wiping	23	MLP, RBF, and SVM	Accuracy: 0.971
[147]	DGAM HA-FA	Acc, Mic, Lig, Bar, Hum, Tem, and Com	(1) sitting, (2) standing, (3) walking, (4) jogging, (5) walking up stairs, (6) walking down stairs, (7) riding a bike, (8) driving a car, (9) riding elevator down, and (10) riding elevator up	41	Ds, and HMM	Accuracy: 0.95 Precision: 0.99 Recall: 0.91

**Table 12 sensors-19-03808-t012:** Datasets.

Id	Dataset
1	Created by Kjærgaard et al. [106]
2	Localization Data for Person Activity [159]
3	Created by Zebin et al. [115]
4	Created by Shoaib et al. [115]
5	WISDM v1.1 [151]
6	ActiveMiles [160]
7	WISDM v2.0 [161]
8	Skoda [162]
9	Daphnet FoG [152]
10	KARD [153]
11	Florence3D [163]
12	Created by Altini et al. [112]
13	Created by John et al. [113]
14	ADL corpus collection [164]
15	Created by Guiry et al. [110]
16	Created by Adelsberger et al. [111]
17	ISI [165]
18	WARD [166]
19	Created by Liu et al. [120]
20	Created by Alam et al. [123]
21	MHEALTH [167]
22	PAMAP2 [168]
23	Created by Chernbumroong at al. [138]
24	Tulum2009 [169]
25	Milan2009 [169]
26	TwoSummer2009 [169]
27	Created by Garcia-Ceja et al. [27]
28	Berkeley MHAD [170]
29	UTD-MHAD [171]
30	Opportunity [172]
31	Created by Bao et al. [121]
32	Created by Arnon [135]
33	Created by Glodek et al. [136]
34	Created by Grokop et al. [141]
35	Created by Zhu et al. [142]
36	Created by Li et al. [144]
37	IXMAS [173]
38	Created by Rad et al. [50]
39	Real [174]
40	Created in eCAALYX project [29]
41	Created by Lester et al. [147]

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
