# Peer review of "Multi-Sensor Fusion for Activity Recognition—A Survey"

_sensors, 2019, doi:10.3390/s19173808_

Round 1
Reviewer 1 Report
In this manuscript, the authors reviewed and compared multiple fusion methods implemented for activity recognition in the literature. Â The work introduced sensors, machine learning techniques, the workflow, Â multi-sensor data fusion of activity recognition in details. Overall, the review presented an integrated, synthesized overview of the current state of human activity recognition. Â However, I still have the following points for the improvement of the work:
1. The following early work about utilizing the sensors of the smartphone in activity recognition need to be discussed in the main text:
Shoaib, M., Bosch, S., Incel, O., Scholten, H., and Havinga, P. (2014). Fusion of smartphone motion sensors for physical activity recognition. Sensors 14, 10146-10176.
He Y, Li Y. Physical activity recognition utilizing the built-in kinematic sensors of a smartphone. International Journal of Distributed Sensor Networks. 2013;9(4):481580.
2. The authors need to describe and discuss more in future research directions.
Author Response
Answer to reviewer remarks 1
Â
Remarks of the reviewer 1
Â
In this manuscript, the authors reviewed and compared multiple fusion methods implemented for activity recognition in the literature. The work introduced sensors, machine learning techniques, the workflow, multi-sensor data fusion of activity recognition in details. Overall, the review presented an integrated, synthesized overview of the current state of human activity recognition. However, I still have the following points for the improvement of the work:
Â
1. The following early work about utilizing the sensors of the smartphone in activity recognition need to be discussed in the main text:
Â
Shoaib, M., Bosch, S., Incel, O., Scholten, H., and Havinga, P. (2014). Fusion of smartphone motion sensors for physical activity recognition. Sensors 14, 10146-10176.
Â
He Y, Li Y. Physical activity recognition utilizing the built-in kinematic sensors of a smartphone. International Journal of Distributed Sensor Networks. 2013;9(4):481580.
Â
Answer to this remark by authors:
Â
We base this survey on the results of the research string of a systematic mapping study, which filters these results with inclusion and exclusion criteria. The work of He et al. did not appear in the research string that we worked with; to modify this research string would imply to redo the entire survey. With respect to the work of Shoaib et al., it did appear in the results of the research string, but we excluded it because Shaoib et al. presented the results of the accuracy of each activity separately in figures as graphs, not showing the actual accuracy: the scale of the figure did not allow knowing the exact values. Therefore, we could not compare the result of Shaoib et al.’s work with others resulting from the research string. However, we decided to describe, in the Survey, the work of Shaiob et al., first, classifying it in Section 5 (page 12, lines 398-402), then, including it in section 6.5 with the corresponding analysis (pages 25-26, lines 853-873); He’s work is also mentioned in the initial classification of Section 1 (page 2, lines 51-52), but is not included in the rest of the Survey. We also notice that we present several methods of hierarchical fusion that are representative of the work of He et al. (pages 14-15, lines 509-545). We also compare and discuss them in Section 6.
Â
2. The authors need to describe and discuss more in future research directions.Â
Â
Answer to this remark by authors:
Â
We have added other promising trends and research directions in Section 6.6. First, we identify and compare the scenarios addressed by the fusion methods, in the new Section 6.4 (page 24, lines 833-851), and then discuss the trend and research gap around those scenarios, in Section 6.6 (page 32, lines 899-910).

Reviewer 2 Report
The paper provide a survey on the recent progress of techniques for human activity recognition, especially for those based on the multi-sensor fusion. The many fusion methods have also compared and examined with their relative merits and replicability, while the trends in the area were also assessed. However, the paper needs to be improved before it can be published on this journal. For this issue, the reviewer has some suggestions and remarks:
1.    The study is less of comprehensive. Most of contents are just simply list the methods and results, while a comprehensive analysis of the reasons why those methods have good or bad performance is missing, this should be improved.
2.    The use of accuracy for evaluation is not enough. The computation complexity and time complexity of these fusion methods are also essential parts to be considered.
3.    This survey mainly relates to the topic of multi-sensor fusion for activity recognition. Despite the many fusion methods list out in the paper, the author did not give a discussion about the scenarios which these methods is suitable. In practice, the demands and requirements for different applications may be quite different.
4.    The content distribution is not balanced. Section 3 and 4 are too verbose, while the discussion part is inadequate, which need to be strengthened. Besides, a clear classification of the literature will be better for reading.
Author Response
Answer to reviewer remarks 2
Â
Remarks of the reviewer 2
Â
The paper provide a survey on the recent progress of techniques for human activity recognition, especially for those based on the multi-sensor fusion. The many fusion methods have also compared and examined with their relative merits and replicability, while the trends in the area were also assessed. However, the paper needs to be improved before it can be published on this journal. For this issue, the reviewer has some suggestions and remarks:
Â
1. The study is less of comprehensive. Most of contents are just simply list the methods and results, while a comprehensive analysis of the reasons why those methods have good or bad performance is missing, this should be improved.
Â
Answer to this remark by authors:
Â
We have made a great effort to identify, analyze and classify the different fusion methods, proposed in the literature, following a categorization proposed by Liggings [1]. The results of this work are fusion methods clearly categorized and compared according to their merits, with their respective debates and trend analysis.Â
Â
One of the findings of the survey is that the fusion methods proposed in the literature do not explain the reasons why they work well for a given set of sensors. On the other hand, Aguileta et al. [2] found that there is no simple reason why a fusion method is good under certain conditions and less good in other conditions. At best, what they accomplished was to predict that a fusion method will work well or not for a combination of features learned by a classifier. This is commented in the Discussion and Trends Section (Section 6.6, page 32, lines 911-922).
Â
[1] Liggins, M.E.; Hall, D.L.; Llinas, J. Handbook of multisensor data fusion: theory and practice, 2009.
Â
[2] Aguileta, A.A.; Brena, R.F.; Mayora, O.; Molino-Minero-Re, E.; Trejo, L.A. Virtual Sensors for Optimal Integration of Human Activity Data. Sensors 2019, 19, 2017.
Â
Â
2. The use of accuracy for evaluation is not enough. The computation complexity and time complexity of these fusion methods are also essential parts to be considered.
Â
Answer to this remark by authors:
Â
We consider that the complexity analysis goes beyond the scope of this survey. In this survey, we have proposed a comparison of the different fusion methods found in the literature, using accuracy metrics, under different perspectives, such as comparing the precision attained whenever a set of heterogeneous sensors is used, against the use of a set of homogeneous ones, independently of the method’s time or computational complexity (see Section 6). Certainly, we do not only use accuracy, we also use precision, recall, sensitivity, specificity, but not a complexity analysis. Additionally, computational or time complexities are almost never reported by authors and very often, they only give a general description of the applied fusion methods which doesn’t allow to perform a correct complexity analysis. Therefore, accuracy and similar metrics provide us with a common ground from which we can understand and compare the various fusion methods.
Â
We acknowledge the importance of time and computational analysis, as mentioned by the reviewer. For example, it would be very valuable to conclude that two fusion methods attained a very similar precision but with very different cost, in terms of asymptotical performance. The computational and time analysis could be of such a great relevance, as to complement our results, in the case, for instance, that the goal was to reduce the resource usage of the hardware being employed, such a CPU/memory or sensors battery. As mentioned before, this is not our main goal in this work.
Â
3. This survey mainly relates to the topic of multi-sensor fusion for activity recognition. Despite the many fusion methods list out in the paper, the author did not give a discussion about the scenarios which these methods is suitable. In practice, the demands and requirements for different applications may be quite different.Â
Â
Â
Answer to this remark by authors:
Â
We identify and compare the scenarios addressed by the fusion methods, in the new Section 6.4 (page 24, lines 833-851), and then discuss the trends and research gaps around those scenarios, in Section 6.6 (page 32, lines 899-910).
Â
Â
4. The content distribution is not balanced. Section 3 and 4 are too verbose, while the discussion part is inadequate, which need to be strengthened. Besides, a clear classification of the literature will be better for reading.Â
Â
Answer to this remark by authors:
Â
We simplified and shortened Section 3. Also, we strengthened the discussions (see response to observation 3).
Â
About the comment on "a clear classification of the literature will be better for reading," we comment that we classify the fusion methods according to the categorization (level of data, level of function and level of decision) suggested by Liggings [1], which is a clear categorization used in the literature. We describe this categorization in section 3.5 and then classify each fusion method in Section 5 following such categorization. Each subsection (5.1, 5.2, and 5.3) of Section 5 presents the mechanics of fusion methods, which justify the category in which they are classified. However, to facilitate the reading of the fusion method classified by the categorization suggested by Liggings, we divided the table of Section 6.5 that summarizes the articles analyzed in this Survey into three tables (9, 10 and 11), which correspond to Liggings categories.
[1] Liggins, M.E.; Hall, D.L.; Llinas, J. Handbook of multisensor data fusion: theory and practice, 2009.

Round 2
Reviewer 2 Report
The typesetting and expression of the paper still needs improvement.
Author Response
Answer to reviewer remarks 2Â
Remark of the reviewer 2
The typesetting and expression of the paper still needs improvement.
Answer to this remark by authors:Â
A thorough terminology, grammatical and spelling revision has been made, in particular, we improved several expressions in the manuscript (see page 1, lines 1-5, 7, 10, 18-20 and 24; page 2, lines 38, 60 and 66; page 3, lines 81, 84 and 92; page 5, lines 118-120; page 9 line 333; page 14 line 505; page 20 lines 743, 754 and 758; page 24 lines 836, 842 and 849; page 25 lines 856-857; and page 32 lines 880, 882, 884, 895 , 898, 902 and 907).
Also, we improved the style of reference citation lists; for example, instead of writing [8] [9] [10], we now write [8-10] (see page 2, lines 31 and 51; page 12, line 401; and page 14, lines 505 and 514).
Besides, we corrected the typing errors of some formulas, such as writing $ xi $ instead of $ x_i $, writing $ fi $ instead of $ f_i $, or writing $ {something_ (wi-wj) ^ 2} $ instead of $ {something} (wi-wj) ^ 2 $Â (see page 7, line 225; page 12, line 415; and page 19, line 706).
Moreover, we fixed the style of the quote marks. Before, we wrote generic quotes (“ "), but now we write open and close quotes (`` '') (see page 2, lines 91-92, 102 and 108; page 13, line 446; page 16, line 591; page 20, lines 746-747; page 22, table title 6 and lines 793, 796 and 799; page 24, legend of Table 7 and lines 836, 838, 840, 842-843, 845-846 and 849-850 and page 25, line 856).
Furthermore, we change the language of Table 8 from Spanish to English (see page 25).
